# Homeostatic and tumourigenic activity of SOX2+ pituitary stem cells is controlled by the LATS/YAP/TAZ cascade

Emily J Lodge[1,2], Alice Santambrogio[1,3], John P Russell[1], Paraskevi Xekouki[1,4], Thomas S Jacques[5], Randy L Johnson[6], Selvam Thavaraj[7], Stefan R Bornstein[2,3], Cynthia Lilian Andoniadou[1,3]*

[1]Centre for Craniofacial and Regenerative Biology, Faculty of Dentistry, Oral & Craniofacial Sciences, King's College London, London, United Kingdom; [2]Division of Diabetes & Nutritional Sciences, Faculty of Life Sciences & Medicine, King's College London, London, United Kingdom; [3]Department of Medicine III, University Hospital Carl Gustav Carus, Technische Universität Dresden, Dresden, Germany; [4]Department of Endocrinology, King's College Hospital NHS Foundation Trust, London, United Kingdom; [5]UCL GOS Institute of Child Health and Great Ormond Street Hospital for Children NHS Foundation Trust, London, United Kingdom; [6]Department of Cancer Biology, The University of Texas, MD Anderson Cancer Center, Houston, United States; [7]Centre for Oral, Clinical and Translational Sciences, Faculty of Dentistry, Oral & Craniofacial Sciences, King's College London, London, United Kingdom

**Abstract** SOX2 positive pituitary stem cells (PSCs) are specified embryonically and persist throughout life, giving rise to all pituitary endocrine lineages. We have previously shown the activation of the STK/LATS/YAP/TAZ signalling cascade in the developing and postnatal mammalian pituitary. Here, we investigate the function of this pathway during pituitary development and in the regulation of the SOX2 cell compartment. Through loss- and gain-of-function genetic approaches, we reveal that restricting YAP/TAZ activation during development is essential for normal organ size and specification from SOX2+ PSCs. Postnatal deletion of LATS kinases and subsequent upregulation of YAP/TAZ leads to uncontrolled clonal expansion of the SOX2+ PSCs and disruption of their differentiation, causing the formation of non-secreting, aggressive pituitary tumours. In contrast, sustained expression of YAP alone results in expansion of SOX2+ PSCs capable of differentiation and devoid of tumourigenic potential. Our findings identify the LATS/YAP/TAZ signalling cascade as an essential component of PSC regulation in normal pituitary physiology and tumourigenesis.
DOI: https://doi.org/10.7554/eLife.43996.001

*For correspondence:
cynthia.andoniadou@kcl.ac.uk

Competing interests: The authors declare that no competing interests exist.

## Introduction

SOX2 is a crucial transcription factor involved in the specification and maintenance of multiple stem cell populations in mammals. Pituitary stem cells express SOX2 and contribute to the generation of new endocrine cells during embryonic development and throughout postnatal life (*Andoniadou et al., 2013*; *Rizzoti et al., 2013*). The pituitary gland is composed of three parts, the anterior, intermediate and posterior lobes (AL, IL and PL, respectively). The AL and IL contain hormone-secreting cells, which are derived from an evagination of the oral ectoderm expressing SOX2, termed Rathke's pouch (RP). SOX2+ cells, both in the embryonic and adult pituitary, can

**eLife digest** The pituitary is a gland inside the head that releases hormones that control major processes in the body including growth, fertility and stress. Diseases of the pituitary gland can prevent the body from producing the appropriate amounts of hormones, and also include tumours.

A population of stem cells in the pituitary known as SOX2 cells divide to make the specialist cells that produce the hormones. This population forms as the pituitary develops in the embryo and continues to contribute new hormone-producing cells throughout life. Signals from inside and outside the gland control how the pituitary develops and maintain the correct balance of different types of cells in the gland in adults.

In 2016, Lodge et al. reported that a cascade of signals known as the Hippo pathway is active in mouse and human pituitary glands, but its role remained unclear. Here, Lodge et al. use genetic approaches to study this signalling pathway in the pituitary of mice.

The results of the experiments show that the Hippo pathway is essential for the pituitary gland to develop normally in mouse embryos. Furthermore, in adult mice the Hippo pathway is required to maintain the population of SOX2 cells in the pituitary and to regulate their cell numbers. Increasing the level of Hippo signalling in mouse embryos and adult mice led to an expansion of SOX2 stem cells that could generate new specialist cell types, but a further increase generated aggressive tumours that originated from the uncontrolled growth of SOX2 cells.

These findings are the first step to understanding how the Hippo pathway works in the pituitary, which may eventually lead to new treatments for tumours and other diseases that affect this gland. The next step towards such treatments will be to carry out further experiments that use drugs to control this pathway and alter the fate of pituitary cells in mice and other animals.

DOI: https://doi.org/10.7554/eLife.43996.002

differentiate into three endocrine cell lineages, which are marked by transcription factors PIT1 (POU1F1) (*Li et al., 1990*), TPIT (TBX19) (*Pulichino et al., 2003*) and SF1 (NR5A1) (*Ingraham et al., 1994*), and differentiate into hormone-secreting cells (somatotrophs, lactotrophs, thyrotrophs, corticotrophs, melanotrophs and gonadotrophs, which express growth hormone, prolactin, thyrotropin, adrenocorticotropin, melanotropin and gonadotropin, respectively). SOX2+ PSCs are highly proliferative during the first 2–3 weeks of life, in concordance with major organ growth, after which they reach a steady low proliferative capacity that contributes to maintain normal homeostasis and physiological adaptation of the pituitary gland (*Levy, 2002*; *Nolan et al., 1998*).

Contrary to other organs, where somatic stem cells are shown to be able to become transformed into cancer stem cells, the roles of SOX2+ PSCs in tumourigenesis remain poorly understood, possibly due to the patchy knowledge of the pathways regulating SOX2+ PSC fate and proliferation. Pituitary tumours are common in the population, representing 10–15% of all intracranial neoplasms (*Bronstein et al., 2011*; *Daly et al., 2006*). Adenomas are the most common adult pituitary tumours, classified into functioning, when they secrete one or more of the pituitary hormones, or non-functioning if they do not secrete hormones. In children, adamantinomatous craniopharyngioma (ACP) is the most common pituitary tumour. Targeting oncogenic beta-catenin in SOX2+ PSCs in the mouse generates clusters of senescent SOX2+ cells that induce tumours resembling ACP in a paracrine manner, that is the tumours do not derive from the targeted SOX2+ PSCs (*Andoniadou et al., 2013*; *Gonzalez-Meljem et al., 2017*). Up to 15% of adenomas and 50% of ACP display aggressive behaviour with invasion of nearby structures including the hypothalamus and visual tracts, associated with significant morbidity and mortality (*Lasolle et al., 2017*). Pituitary carcinomas exhibiting metastasis are rare but can develop from benign tumours (*Veldhuis, 2013*; *Pernicone et al., 1997*; *Heaney, 2014*). Whether SOX2+ cells can cell autonomously contribute to pituitary neoplasia has not been hitherto demonstrated.

The Hippo pathway controls stem cell proliferation and tumourigenesis in several organs such as in the liver (*Zhou et al., 2009*; *Lu et al., 2018*), intestines (*Zhou et al., 2011*) and lung (*Lin et al., 2015*; *Nantie et al., 2018*). In the core phosphorylation cascade, STK3/4 kinases phosphorylate and activate LATS1/2 serine/threonine-protein kinases, which in turn phosphorylate co-activators Yes-associated protein (YAP1, a.k.a. YAP) and WW domain-containing transcription regulator protein 1

(WWTR1, a.k.a. TAZ) that are subsequently inactivated through degradation and cytoplasmic retention (*Meng et al., 2016*). Active YAP/TAZ associate with TEAD transcription factors, promoting the transcription of target genes such as *Cyr61* and *Ctgf* (*Zhao et al., 2008*; *Zhang et al., 2009*; *Zhou et al., 2016*). YAP/TAZ have been shown to promote proliferation and the stem cell state in several organs, and can also lead to transformation and tumour initiation when overexpressed (*Camargo et al., 2007*; *Schlegelmilch et al., 2011*; *Dong et al., 2007*). The involvement of YAP/TAZ in the function of tissue-specific SOX2+ stem cells during development and homeostasis has not been shown. We previously reported strong nuclear localisation of YAP and TAZ exclusively in SOX2+ stem cells of developing Rathke's pouch and the postnatal anterior pituitary of mice and humans, and enhanced expression in human pituitary tumours composed of uncommitted cells, including ACPs and null-cell adenomas (*Lodge et al., 2016*; *Xekouki et al., 2019*), which do not express any of the lineage transcription factors PIT1, TPIT or SF1. In these populations we detected phosphorylation of YAP at serine 127 (S127) indicating LATS kinase activity. Together these point to a possible function for LATS/YAP/TAZ in normal pituitary stem cells and during tumourigenesis. Here, we have combined genetic and molecular approaches to reveal that deregulation of the pathway can promote and maintain the SOX2+ PSC fate under physiological conditions and that major disruption of this axis transforms SOX2+ PSCs into cancer-initiating cells giving rise to aggressive tumours.

## Results

### Sustained conditional expression of YAP during development promotes SOX2+ PSC fate

To determine if YAP and TAZ function during embryonic development of the pituitary, we used genetic approaches to perform gain- and loss-of-function experiments. We first expressed a constitutive active form of YAP(S127A) using the *Hesx1-Cre* driver, which drives *Cre* expression in Rathke's pouch (RP) and the hypothalamic primordium from 9.5dpc, regulated by administration of doxycycline through the reverse tetracycline-dependent transactivator (rtTA) system ($R26^{rtTA/+}$; see Materials and methods for details, Scheme *Figure 1A*). Analyses were restricted to embryonic time points. As expected, we confirmed accumulation of total YAP protein but not of TAZ or pYAP(S127), throughout the developing pituitary and hypothalamus of $Hesx1^{Cre/+};R26^{rtTA/+};Col1a1^{tetO-Yap/+}$ (hereafter YAP-TetO) embryos at 15.5dpc, but not of *Cre*-negative controls (*Figure 1B*, *Figure 1—figure supplement 1A*). Likewise, the YAP downstream target *Cyr61* (*Figure 1B*) was also upregulated. Morphologically, YAP-TetO mutants displayed a dysplastic anterior pituitary, which was more medially compacted and lacked a central lumen, making it difficult to distinguish between the developing anterior and intermediate lobes (*Figure 1C*). Immunofluorescence staining against SOX2 at 15.5dpc demonstrated loss of SOX2 in the most lateral regions of control pituitaries (arrows in *Figure 1C*), where cells are undergoing commitment; yet mutant pituitaries had abundant SOX2 positive cells in the most lateral regions (arrowheads in *Figure 1C*). Immunostaining for LHX3, which is expressed in the developing anterior pituitary (*Sheng et al., 1996*), was used to demarcate AL and IL tissue. Staining using antibodies against lineage markers PIT1, TPIT and SF1 revealed a concomitant reduction in committed cell lineages throughout the gland (*Figure 1D*; PIT1 0.35% in mutants compared with 30.21% in controls (Student's t-test p<0.0001, n = 3 for each genotype), TPIT 1.03% in mutants compared with 9.81% in controls (Student's t-test p=0.0012, n = 3 for each genotype), SF1 0.34% in mutants compared with 4.14% in controls (Student's t-test p=0.0021, n = 3 for each genotype)). We therefore conclude that sustained activation of YAP prevents lineage commitment and is sufficient to maintain the progenitor state during embryonic development.

We did not obtain any live $Hesx1^{Cre/+};R26^{rtTA/+};Col1a1^{tetO-Yap/+}$ pups at birth when treated with doxycycline from 5.5dpc (n = 5 litters). To bypass the embryonic lethality of these early inductions, we commenced doxycycline treatment from 9.5dpc, the onset of RP formation (*Figure 1—figure supplement 1B*).

$Hesx1^{Cre/+};R26^{rtTA/+};Col1a1^{tetO-Yap/+}$ pups were viable and were maintained on doxycycline until P24, at which point the experimental end point was reached due to excessive weight loss and animals had to be culled following UK Home Office Regulations. Histological analyses of pituitaries revealed multiple anterior lobe cysts per gland, localising predominantly in the ventral AL (n = 4)

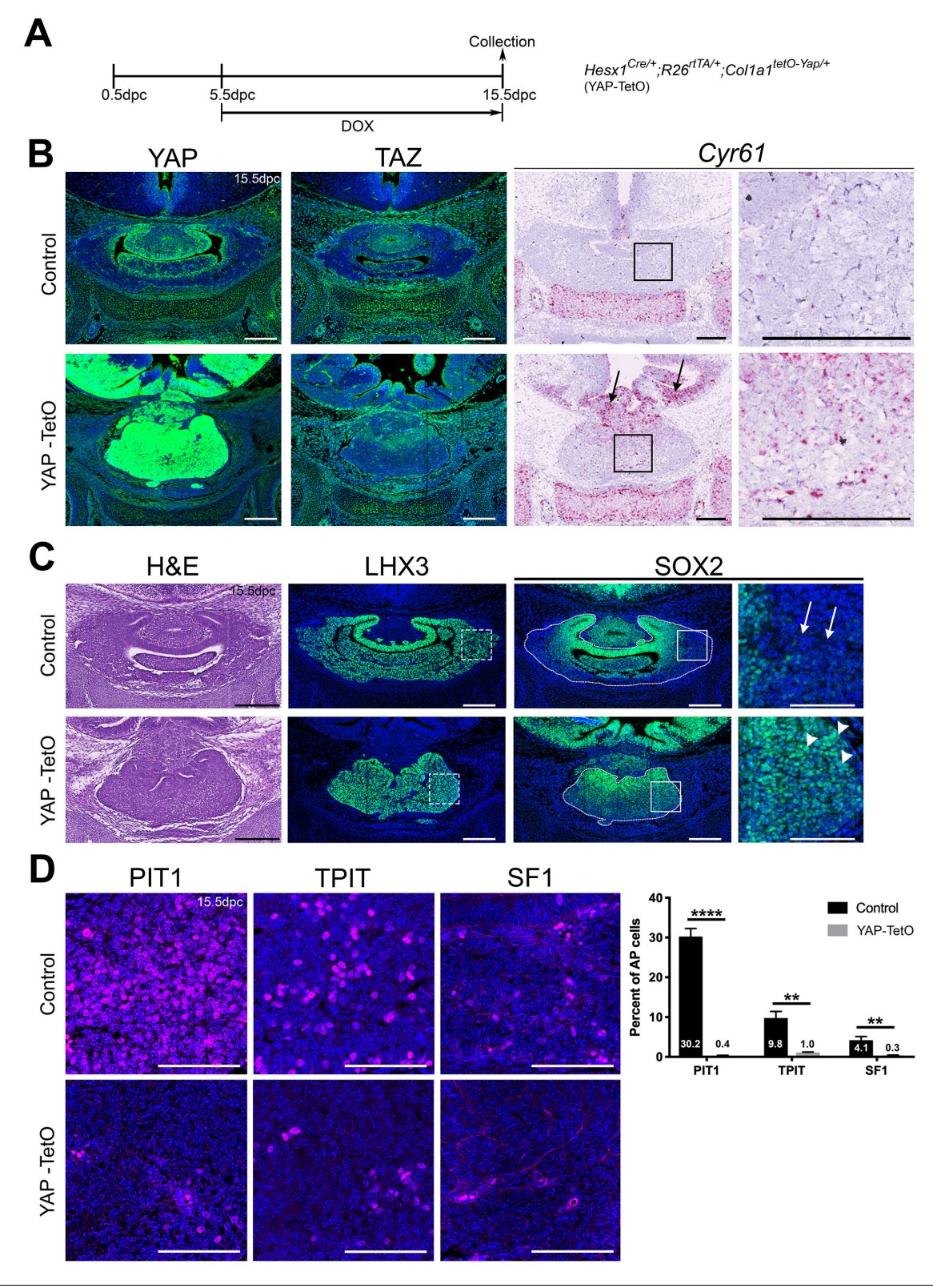

**Figure 1.** Regulation of YAP is required for normal morphogenesis and lineage commitment during pituitary development. (**A**) Schematic outlining the time course of doxycycline (DOX) treatment administered to pregnant dams from $Hesx1^{Cre/+}$ x $R26^{rtTA/rtTA}$;$Col1a1^{tetO-Yap/tetO-Yap}$ crosses for the embryonic induction of YAP(S127A) expression in $Hesx1^{Cre/+}$;$R26^{rtTA/+}$;$Col1a1^{tetO-Yap/+}$ (YAP-TetO) mutant embryos as well as controls that do not express YAP(S127A) ($Hesx1^{+/+}$;$R26^{rtTA/+}$;$Col1a1^{tetO-Yap/+}$ controls shown here). (**B**) Immunofluorescence staining against YAP and TAZ on frontal pituitary

*Figure 1 continued on next page*

Figure 1 continued

sections at 15.5dpc confirms accumulation of YAP protein in YAP-TetO compared to control sections, but no increase in TAZ levels. RNAscope mRNA in situ hybridisation against the YAP/TAZ target *Cyr61* confirms an increase in transcripts in the anterior pituitary as well as the hypothalamus where the Cre is also active (arrows). (C) Haematoxylin and eosin staining of frontal pituitary sections from 15.5dpc control and YAP-TetO embryos showing pituitary dysmorphology in mutants. Immunofluorescence staining for LHX3 to mark anterior pituitary tissue and SOX2 to mark pituitary progenitors shows the persistence of SOX2 protein in lateral regions of the gland in YAP-TetO mutants (arrowheads) when they have lost SOX2 expression in controls (arrows) (magnified boxed region in SOX2, corresponding to dashed box in LHX3). (D) Immunofluorescence staining for lineage-committed progenitor markers PIT1, TPIT and SF1 reveals very few cells expressing commitment markers in YAP-TetO compared to control. Graph showing quantification of committed cells of the three anterior pituitary endocrine lineages, positive for PIT1, TPIT and SF1, as a percentage of total nuclei of $Hesx1^{+/+};R26^{rtTA/+};Col1a1^{tetO-Yap/+}$ control and $Hesx1^{Cre/+};R26^{rtTA/+};Col1a1^{tetO-Yap/+}$ (YAP-TetO) mutant pituitaries at 15.5dpc (Student's *t*-test; PIT1: p<0.0001 (****), TPIT: p=0.0012 (**), SF1: p=0.0021 (**)). Scale bars 100 μm, 50 μm in magnified boxed regions in C. See also *Figure 1—figure supplements 1 and 2*.

DOI: https://doi.org/10.7554/eLife.43996.003

The following figure supplements are available for figure 1:

**Figure supplement 1.** Regulation of YAP and TAZ during pituitary development.

DOI: https://doi.org/10.7554/eLife.43996.004

**Figure supplement 2.** Regulation of YAP and TAZ during pituitary development.

DOI: https://doi.org/10.7554/eLife.43996.005

(*Figure 1—figure supplement 1C*). These structures developed in YAP-accumulating regions and were lined by SOX2+ cells (*Figure 1—figure supplement 1D*). The proportion of SOX2+ cells throughout the AL was increased, as was the percentage of SF1+ cells, whereas PIT1+ cell numbers were significantly decreased and differentiated cells of the TPIT lineage, identified by ACTH antibody staining, were unaffected (*Figure 1—figure supplement 1E*). The total number of cycling Ki-67+ cells showed a trend towards a decrease in $Hesx1^{Cre/+};R26^{rtTA/+};Col1a1^{tetO-Yap/+}$ mutants relative to controls, which did not reach significance (*Figure 1—figure supplement 1F*). The cystic structures observed in $Hesx1^{Cre/+};R26^{rtTA/+};Col1a1^{tetO-Yap/+}$ mutants were reminiscent of Rathke's cleft cyst (RCC), which is a benign developmental anomaly of the pituitary characterised by the presence of ciliated and secretory cells, expression of cytokeratins and frequent expression of p63. Immunostaining revealed that cysts were lined by cytokeratin+ cells using the AE1/AE3 pan-cytokeratin cocktail and basal cells were positive for nuclear p63 in $Hesx1^{Cre/+};R26^{rtTA/+};Col1a1^{tetO-Yap/+}$ mutant pituitaries (*Figure 1—figure supplement 1G*). Staining using antibodies against ARL13B and Acetylated α-Tubulin (Lys40) marking cilia, revealed multi-ciliated cells along the cyst lining (*Figure 1—figure supplement 1H*). Combined staining using Alcian Blue and the Periodic Acid-Schiff technique (AB/PAS) to recognise mucins, detected royal blue-stained mucous cells lining the cysts (*Figure 1—figure supplement 1H*). Taken together, we conclude that sustained activation of YAP during embryonic and postnatal pituitary development, promotes maintenance and abnormal expansion of SOX2+ epithelia during development, resulting in the formation of cysts that resemble RCC.

Next, we generated embryos null for TAZ and conditionally lacking YAP in the *Hesx1* expression domain (*Figure 1—figure supplement 2A–E*). $Hesx1^{Cre/+};Yap^{fl/fl};Taz^{-/-}$ double mutants were obtained at expected ratios during embryonic stages until 15.5dpc, however the majority of $Taz^{-/-}$ mutants with or without compound *Yap* deletions showed lethality at later embryonic and early postnatal stages (*Tian et al., 2007*) (*Supplementary file 1*). The developing pituitary gland of $Hesx1^{Cre/+};Yap^{fl/fl};Taz^{-/-}$ double mutants appeared largely normal at 13.5dpc by histology (*Figure 1—figure supplement 2A*). Immunostaining against SOX2 to mark embryonic progenitors and postnatal stem cells did not reveal differences in the spatial distribution of SOX2+ cells between double mutants compared to controls ($Hesx1^{+/+};Yap^{fl/fl};Taz^{+/+}$ and $Hesx1^{+/+};Yap^{fl/fl};Taz^{+/-}$) at 13.5dpc, 16.0dpc (*Figure 1—figure supplement 2B*) or P28, even in regions devoid of both TAZ and active YAP (*Figure 1—figure supplement 2C,D*). This suggests that YAP/TAZ are not required for SOX2+ cell specification or survival. Likewise, analysis of commitment markers PIT1 and SF1 as well as ACTH to identify cells of the TPIT lineage, did not show any differences between genotypes (*Figure 1—figure supplement 2E*). Together, these data suggest there is no critical requirement for YAP and TAZ during development for the specification of SOX2+ cells or lineage commitment, but that YAP functions to promote the SOX2 cell identity.

# LATS, but not STK, kinases are required for normal pituitary development and differentiation

Since sustained activation of YAP led to an embryonic phenotype, we reasoned that YAP/TAZ need to be regulated during embryonic development. To determine if STK and LATS kinases are important in YAP/TAZ regulation we carried out genetic deletions in the pituitary.

Conditional deletion of *Stk3* and *Stk4* (also called *Mst2* and *Mst1*) in *Hesx1$^{Cre/+}$;Stk3$^{fl/fl}$;Stk4$^{fl/fl}$* embryos did not lead to a pituitary phenotype (*Figure 2—figure supplement 1*). A reduction of over 75% in total STK3/4 proteins in mutants was confirmed by western blot on total lysates from *Hesx1$^{+/+}$;Stk3$^{fl/fl}$;Stk4$^{fl/fl}$* controls and *Hesx1$^{Cre/+}$;Stk3$^{fl/fl}$;Stk4$^{fl/fl}$* mutants (*Figure 2—figure supplement 1B*). Mutant pituitaries were macroscopically normal at birth (*Figure 2—figure supplement 1A*), and showed comparable expression patterns of TAZ, YAP, pYAP to controls lacking *Cre*, without distinct accumulation of YAP or TAZ (*Figure 2—figure supplement 1C*). The distribution of SOX2+ cells was comparable between mutants and controls (*Figure 2—figure supplement 1C*). Normal lineage commitment was evident by immunofluorescence staining for PIT1, TPIT and SF1 at P10 (*Figure 2—figure supplement 1D*). Mutant animals remained healthy and fertile until P70, at which point pituitaries appeared histologically normal (*Figure 2—figure supplement 1E*). Since deletion of *Stk3/4* at embryonic stages does not affect embryonic or postnatal pituitary development, we conclude these kinases are not critical for YAP/TAZ regulation in the pituitary.

We next focused on perturbing LATS kinase function, as we have previously shown strong expression of *Lats1* in the developing pituitary and postnatal kinase activity in SOX2+ stem cells (*Lodge et al., 2016*). However, *Hesx1$^{Cre/+}$;Lats1$^{fl/fl}$* embryos showed unaffected pituitary development and normal localisation and levels of YAP and TAZ as assessed by immunofluorescence (*Figure 2—figure supplement 2A,B*) when compared with controls. mRNA in situ hybridisation against *Lats2* at P2, revealed abundant *Lats2* transcripts upon conditional deletion of *Lats1*, suggesting a compensatory upregulation of *Lats2* in the absence of LATS1 (*Figure 2—figure supplement 2C*), similar to previous reports of elevated YAP/TAZ signalling inducing *Lats2* expression (*Moroishi et al., 2015*).

To overcome potential functional redundancy, we deleted both *Lats1* and *Lats2* in RP. Deletion of *Lats2* alone (*Hesx1$^{Cre/+}$;Lats2$^{fl/fl}$*), did not reveal any developmental morphological anomalies (*Figure 2—figure supplement 2D*) and pups were identified at normal Mendelian proportions (*Supplementary file 2*). Similarly, deletion of any three out of four *Lats* alleles did not affect pituitary development and were identified at normal ratios, similar to other tissues (*Lavado et al., 2018*). Homozygous *Hesx1$^{Cre/+}$;Lats1$^{fl/fl}$;Lats2$^{fl/fl}$* mutants were identified at embryonic stages at reduced Mendelian ratios and were absent at P0-P2, suggesting embryonic and perinatal lethality (*Supplementary file 2*).

Haematoxylin/eosin staining of the developing pituitary gland in *Hesx1$^{Cre/+}$;Lats1$^{fl/fl}$;Lats2$^{fl/fl}$* mutants revealed overgrowth of RP by 13.5dpc compared to controls lacking *Cre* (*Figure 2A*, n = 4). Total TAZ and YAP proteins accumulated throughout the developing gland in double mutants (arrowheads) but only in the SOX2+ periluminal epithelium of controls (arrows). The same regions showed a marked reduction in pYAP-S127 staining, which is observed in SOX2+ cells of the control (*Figure 2A*). These findings are in line with LATS1/2 normally regulating YAP and TAZ in the pituitary and demonstrate successful deletion in RP. The mutant pituitary was highly proliferative (*Figure 2B*, *Figure 2—figure supplement 2F*; Ki-67 index average 47.42% ± 1.73 SEM in control versus 76.04% ± 9.11 SEM in the double mutant, p=0.0067, Student's *t*-test) and the majority of cells expressed SOX2 (*Figure 2A,C*) but not SOX9 (*Figure 2B*, *Figure 2—figure supplement 2F*).

By 15.5dpc the pituitary was grossly enlarged and exerting a mass effect on the brain, had cysts and displayed areas of necrosis (asterisks *Figure 2*, *Figure 2—figure supplement 2E*, n = 5). Staining for Endomucin to mark blood vessels revealed poor vascularisation in *Hesx1$^{Cre/+}$;Lats1$^{fl/fl}$;Lats2$^{fl/fl}$* mutants compared to the ample capillaries seen in the control (*Figure 2C*), which may account for the necrosis. This could be due to a direct inhibition of vascularisation or a consequence of the rapid growth of this embryonic tumour. We frequently observed ectopic residual pituitary tissue at more caudal levels, reaching the oral epithelium and likely interfering with appropriate fusion of the sphenoid, similar to other phenotypes involving pituitary enlargement (arrows *Figure 2C*) (*Andoniadou et al., 2012*; *Sajedi et al., 2008*; *Gaston-Massuet et al., 2008*). Immunofluorescence to detect active (non-phosphorylated) YAP revealed abundant staining throughout the pituitary at

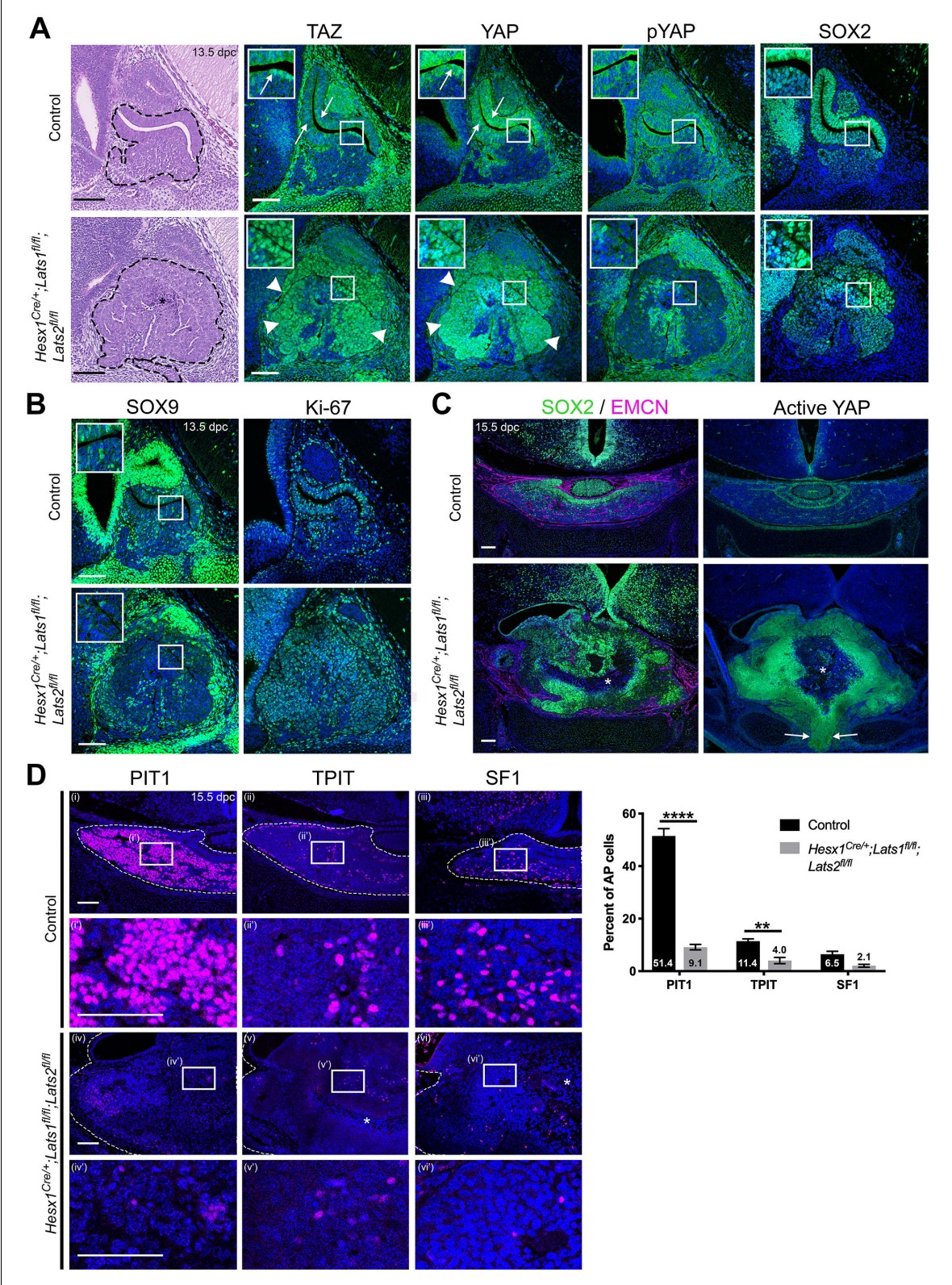

**Figure 2.** Pituitary-specific deletion of *Lats1* and *Lats2* during development leads to pituitary overgrowth and defects in lineage commitment. (**A**) Haematoxylin and eosin staining on sagittal sections from *Hesx1*^Cre/+;*Lats1*^fl/fl;*Lats2*^fl/fl (mutant) and *Hesx1*^+/+;*Lats1*^fl/fl;*Lats2*^fl/fl (control) embryos at 13.5dpc reveals anterior pituitary dysmorphology and overgrowth in mutants (dashed outline). Immunofluorescence staining for TAZ, YAP and pYAP reveals accumulation of TAZ and YAP in overgrown mutant tissue (arrowheads, normal epithelial expression indicated by arrows in control) and lack of
*Figure 2 continued on next page*

*Figure 2 continued*

staining for pYAP (S127). Immunofluorescence for SOX2 shows the presence of SOX2+ progenitors throughout the abnormal tissue in mutants. (B) Immunofluorescence staining for late progenitor marker SOX9 shows localisation in few cells of the pituitary of mutants at 13.5dpc. Immunofluorescence staining for Ki-67 indicates cycling cells throughout the mutant pituitary. (C) Immunofluorescence staining for SOX2 and Endomucin (EMCN) on frontal pituitary sections at 15.5dpc shows expansion of the SOX2+ progenitor compartment compared to controls and a reduction in vasculature marked by Endomucin. Immunofluorescence for non-phosphorylated (Active) YAP shows strong expression throughout the mutant gland compared to the control. Areas of necrosis in mutant tissue indicated by asterisks. Ventral overgrowth extending into the oral cavity between the condensing sphenoid bone indicated by arrows. (D) Immunofluorescence staining for lineage-committed progenitor markers PIT1, TPIT and SF1 reveals only sporadic cells expressing commitment markers in $Hesx1^{Cre/+};Lats1^{fl/fl};Lats2^{fl/fl}$ mutants compared to controls. Boxes showing magnified regions. Dashed lines demarcate anterior pituitary tissue. Graph showing quantification of committed cells of the three anterior pituitary endocrine lineages, positive for PIT1, TPIT and SF1, as a percentage of total nuclei of $Hesx1^{+/+};Lats1^{fl/fl};Lats2^{fl/fl}$ control and $Hesx1^{Cre/+};Lats1^{fl/fl};Lats2^{fl/fl}$ mutant pituitaries at 15.5dpc (Student's *t*-test; PIT1: p<0.0001 (****), TPIT: p=0.007 (**), SF1: p>0.05). Scale bars 100 µm. See also *Figure 2—figure supplement 2*.

DOI: https://doi.org/10.7554/eLife.43996.006

The following figure supplements are available for figure 2:

**Figure supplement 1.** Pituitary-specific loss of *Stk3* and *Stk4* does not affect SOX2 cell specification or lineage commitment.

DOI: https://doi.org/10.7554/eLife.43996.007

**Figure supplement 2.** Isolated deletions of *Lats1* or *Lats2* in the pituitary do not affect development.

DOI: https://doi.org/10.7554/eLife.43996.008

15.5dpc, compared to the control where active YAP localises in the SOX2 epithelium (*Figure 2C*). Immunofluorescence using specific antibodies against lineage commitment markers PIT1, TPIT and SF1 at 15.5dpc revealed very few cells expressing PIT1, TPIT and SF1 in the double mutant (*Figure 2D*; PIT1 9.14% in mutants compared with 51.4% in controls (Student's *t*-test p<0.0001); TPIT 4.0% in mutants compared with 11.4% in controls (Student's *t*-test p<0.007); SF1 2.1% in mutants compared with 6.5% in controls (Student's *t*-test p>0.05) n = 3 mutants and five controls), suggesting failure to commit into the three lineages. These data suggest that the LATS/YAP/TAZ axis is required for normal embryonic development of the anterior pituitary and that LATS1/2 kinases control proliferation of SOX2+ progenitors and their progression into the three committed lineages.

## Loss of LATS kinases results in carcinoma-like murine tumours

Postnatal analysis of $Hesx1^{Cre/+};Lats1^{fl/fl}$ pituitaries revealed that by P56, despite developing normally during the embryonic period, all glands examined exhibited lesions of abnormal morphology consisting of overgrowths, densely packed nuclei and loss of normal acinar architecture (n = 15). To minimise the likely redundancy by LATS2 seen at embryonic stages, we generated *Lats1* mutants additionally haploinsufficient for *Lats2* ($Hesx1^{Cre/+};Lats1^{fl/fl};Lats2^{fl/+}$). These pituitaries also developed identifiable lesions accumulating YAP and TAZ (*Figure 3—figure supplement 1A*), which were observed at earlier time points (P21 n = 4), the earliest being 10 days, indicating increased severity. The number of lesions observed per animal was similar between the two models at P56 (3–8 per animal). Deletion of *Lats2* alone ($Hesx1^{Cre/+};Lats2^{fl/fl}$), which is barely expressed in the wild type pituitary, did not result in any defects (*Figure 3—figure supplement 1B*). We focused on the $Hesx1^{Cre/+};Lats1^{fl/fl};Lats2^{fl/+}$ double mutants for further analyses.

Histological examination of $Hesx1^{Cre/+};Lats1^{fl/fl};Lats2^{fl/+}$ pituitaries confirmed the abnormal lesions were tumours, characterised by frequent mitoses, focal necrosis, and a focal squamous differentiation, as well as the occasional presence of cysts (*Figure 3A*). These lesions were identical to those in $Hesx1^{Cre/+};Lats1^{fl/fl}$ pituitaries (not shown). These tumours accumulated YAP/TAZ and upregulated expression of targets *Cyr61* and *Ctgf* (*Figure 3B*), confirming the validity of the genetic manipulation (*Figure 3B*). Tumours were also frequently observed in the anterior and intermediate lobe (*Figure 3—figure supplement 1C*). Analysis of proliferation by Ki-67 immunostaining revealed an elevated mitotic index of 7–28% in tumours (mean 15.46, SEM ±2.74), compared to 2.97% (SEM ±1.2) mean in control pituitaries not carrying the *Lats1* deletion (*Figure 3C*).

In keeping with the morphological evidence of epithelial differentiation (*Figure 3A*), the tumours were positive for cytokeratins using AE1/AE3 (multiple keratin cocktail) (*Figure 3—figure supplement 1E*). Furthermore, the tumours showed focal morphological evidence of squamous differentiation and showed positive nuclear p63 staining, frequently expressed in squamous carcinomas

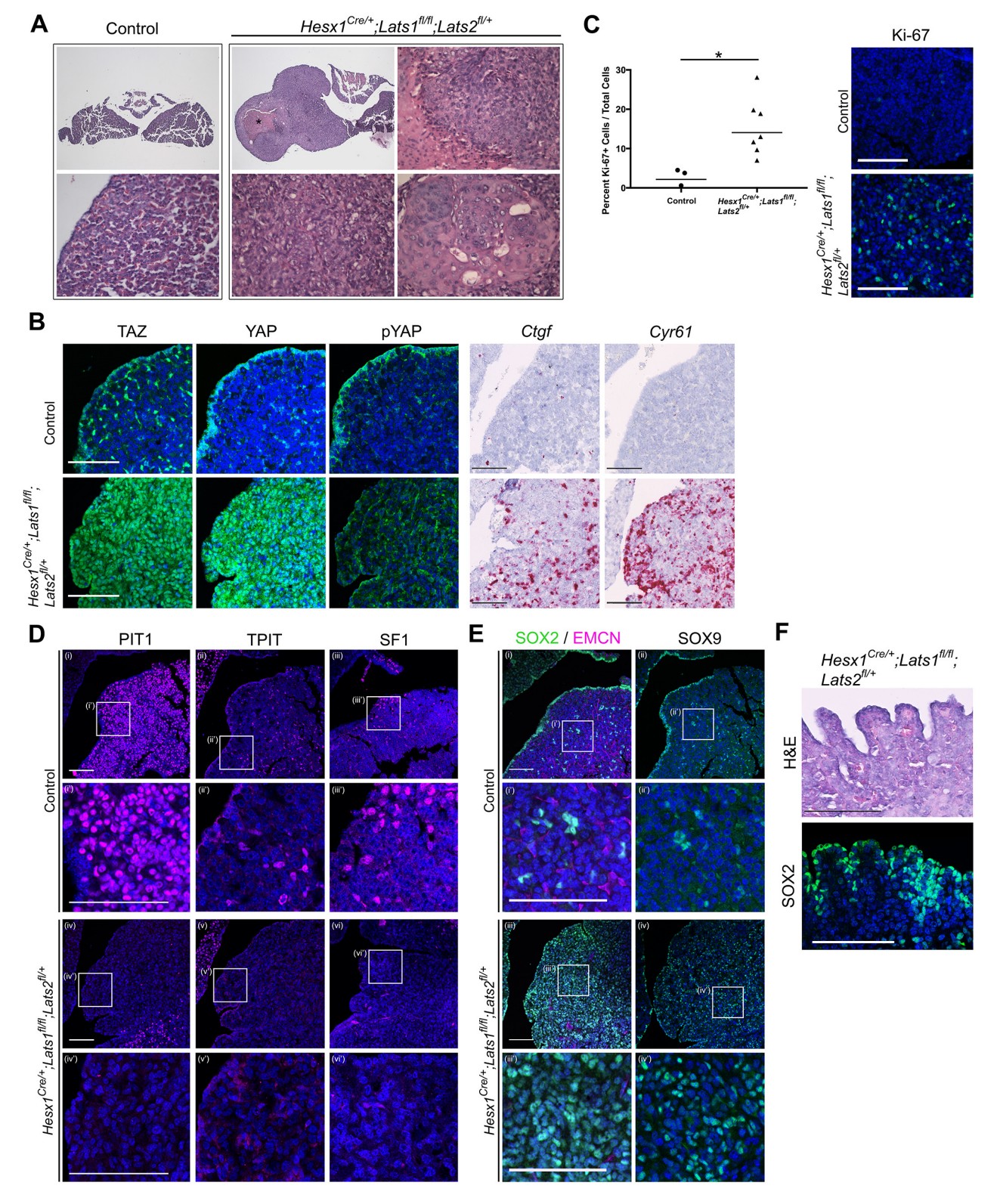

**Figure 3.** Pituitary specific loss of *Lats1* leads to tumour formation. (**A**) Haematoxylin and eosin staining of frontal sections from *Hesx1^{Cre/+};Lats1^{fl/fl}; Lats2^{fl/+}* (mutant) and control pituitaries at P56 demonstrates overgrown tumourigenic regions in mutants. These show focal necrosis, cysts and a squamous morphology (magnified regions) not seen in controls. Asterisk indicates necrosis. (**B**) Immunofluorescence staining for TAZ, YAP and pYAP (S127) show accumulation of TAZ and YAP but not pYAP in the mutant but not in the control. RNAscope mRNA in situ hybridisation against YAP/TAZ

*Figure 3 continued on next page*

*Figure 3 continued*

targets *Ctgf* and *Cyr61* reveals an increase in transcripts on mutant tissue compared to control. (C) Graph of the proliferation index in control and mutant samples at P56 shows a significant increase in cycling cells in the *Hesx1^{Cre/+};Lats1^{fl/fl};Lats2^{fl/+}* mutant pituitaries compared to controls (control percentage Ki-67: 2.967 ± 1.2 SEM, n = 3; mutant: 15.46 ± 2.74 n = 7. p=0.0217 (*), two-tailed *t*-test). Images show representative examples of Ki-67 immunofluorescence staining. (D) Immunofluorescence staining for lineage-committed progenitor markers PIT1, TPIT and SF1 shows the near absence of committed cells in tumours. (E) Immunofluorescence staining for pituitary stem cell markers SOX2 and SOX9 reveal that tumour lesions have abundant positive cells compared to the control, while Endomucin (EMCN) staining shows poor vascularisation. (F) The marginal zone epithelium of *Hesx1^{Cre/+};Lats1^{fl/fl};Lats2^{fl/+}* mutant pituitaries develops invaginations as seen by haematoxylin and eosin staining. Immunofluorescence staining against SOX2 shows the maintenance of a single-layered epithelium. Scale bars 100 µm. Boxes indicate magnified regions. See also *Figure 3—figure supplement 1*.

DOI: https://doi.org/10.7554/eLife.43996.009

The following figure supplement is available for figure 3:

**Figure supplement 1.** Analysis of tumourigenic lesions in postnatal pituitaries following pituitary-specific deletion of *Lats1*.

DOI: https://doi.org/10.7554/eLife.43996.010

(*Figure 3—figure supplement 1E*). In contrast, the tumours did not show immunohistochemical evidence of adenomas, that is, they were negative for neuroendocrine markers, which all types of adenomas are typically positive for: the neuroendocrine marker synaptophysin and neuron-specific enolase (*Figure 3—figure supplement 1F*). The lesions were also negative for chromogranin A, a neuroendocrine granule marker often expressed in clinically non-functioning pituitary adenomas. Tumours were also negative for vimentin, expressed by spindle cell oncocytoma, an uncommitted posterior pituitary tumour (*Figure 3—figure supplement 1F*). Moreover, immunostaining against PIT1, TPIT and SF1 showed only sparse positive cells within the lesions, suggesting lack of commitment into endocrine precursors and supporting the undifferentiated nature of the tumour cells (*Figure 3D*). Consistent with a tumourigenic phenotype, and role for LATS1 genomic stabilisation (*Pefani et al., 2014*), staining for gamma-H2A.X detected elevated DNA damage in cells of the mutant pituitaries compared with controls (*Figure 3—figure supplement 1D*). The absence of adenoma or oncocytoma markers together with the histological appearance, observation of focal necrosis and a high mitotic index support the features of squamous carcinoma.

## SOX2+ cells are the cell of origin of the tumours

Tumour regions were mostly composed of SOX2 positive cells, a sub-population of which also expressed SOX9 (*Figure 3E*, *Figure 3—figure supplement 1A*; 85–97% of cells, 7 tumours across four pituitaries). Close examination of the marginal zone epithelium, a major SOX2+ stem cell niche of the pituitary, revealed a frequent 'ruffling' resembling crypts, likely generated through over-proliferation of the epithelial stem cell compartment (*Figure 3F*). To determine if the cell of origin of the tumourigenic lesions is a deregulated SOX2+ stem cell, we carried our specific deletion of LATS1/2 in postnatal SOX2+ cells using the tamoxifen-inducible *Sox2-CreERT2* driver, combined with conditional expression of membrane-GFP in targeted cells (*Sox2^{CreERT2/+};Lats1^{fl/fl};Lats2^{fl/+};R26^{mTmG/+}*).

Tamoxifen induction at P5 or P21, led to abnormal lesions in the anterior pituitary within three months in all cases. We focused our analyses on inductions performed at P5, from which time point all animals developed lesions by P35 (*Figure 4A*). Similar to observations in *Hesx1^{Cre/+};Lats1^{fl/fl};Lats2^{fl/+}* animals, these areas strongly accumulated YAP and TAZ (*Figure 4B*), activated expression of targets *Cyr61* and *Ctgf*, displayed ruffling of the AL epithelium (*Figure 4C*, *Figure 4—figure supplement 1E*) and lacked lineage commitment markers (*Figure 4D*, *Figure 4—figure supplement 1A*). These lesions showed a similar marker profile to *Hesx1-Cre*-targeted tumours, with positive p63 and AE1/AE3 staining (*Figure 4—figure supplement 1B*). Lineage tracing confirmed expression of membrane GFP in tumourigenic lesions, characterised by the accumulation of YAP and expansion of SOX2+ cells, suggesting they were solely derived from SOX2+ cells (*Figure 4E*, *Figure 4—figure supplement 1C*). Taken together, our data support that LATS kinase activity is required to regulate the pituitary stem cell compartment. Loss of LATS1 is sufficient to drive deregulation of SOX2+ pituitary stem cells, generating highly proliferative non-functioning tumours with features of carcinomas.

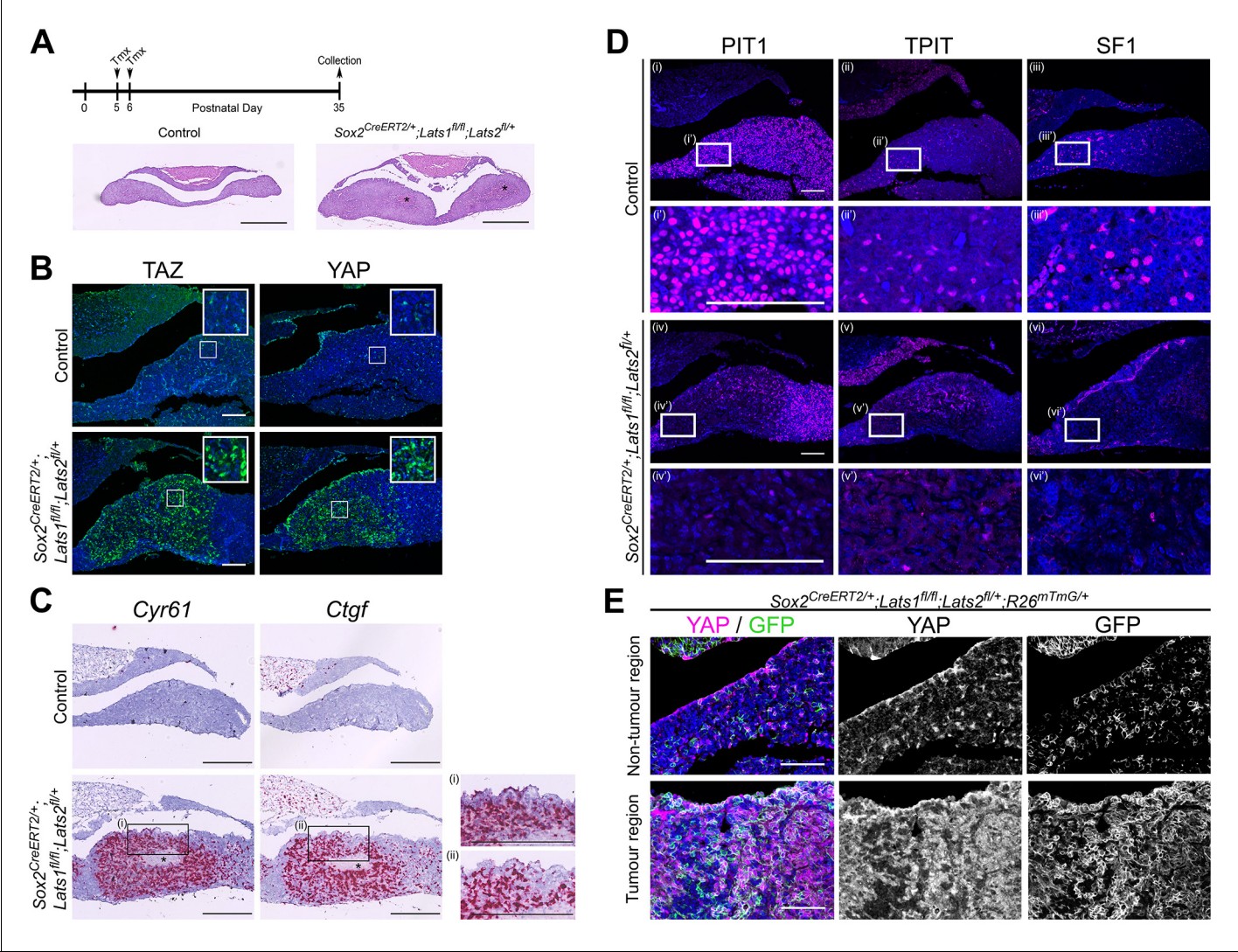

**Figure 4.** SOX2+ pituitary stem cells are the cell-of-origin of tumours generated in the absence of *Lats1*. (**A**) Schematic outlining the experimental time line of inductions in *Sox2^{CreERT2/+};Lats1^{fl/fl};Lats2^{fl/+}* (mutant) and *Sox2^{+/+};Lats1^{fl/fl};Lats2^{fl/+}* (control) animals. Representative images of haematoxylin and eosin staining of frontal sections of control and mutant pituitaries at P35, revealing a hyperplastic anterior pituitary in the mutant with areas of necrosis (asterisks). (**B**) Immunofluorescence staining reveals tumourigenic lesions in *Sox2^{CreERT2/+};Lats1^{fl/fl};Lats2^{fl/+}* that display increased levels of TAZ and YAP staining compared to the control. (**C**) RNAscope mRNA in situ hybridisation against *Ctgf* and *Cyr61* shows elevated transcripts in tumourigenic lesions. Insets (i) and (ii) show invaginations in the epithelium of the mutant. (**D**) Immunofluorescence staining for lineage-committed progenitor markers PIT1, TPIT and SF1 showing a reduction in staining in tumourigenic lesions compared to control pituitaries. (**E**) Lineage tracing of SOX2+ cells in *Sox2^{CreERT2/+};Lats1^{fl/fl};Lats2^{fl/+}R26^{mTmG/+}* reveals that tumour regions accumulating YAP as seen by immunofluorescence, are composed of GFP+ cells at P35. Scale bars 500 μm in A; 100 μm in B, D, E; 250 μm in C. See also *Figure 4—figure supplement 1*.

DOI: https://doi.org/10.7554/eLife.43996.011

The following figure supplement is available for figure 4:

**Figure supplement 1.** Analysis of tumourigenic lesions in postnatal pituitaries following SOX2-specific deletion of *Lats1*.
DOI: https://doi.org/10.7554/eLife.43996.012

## YAP expression is sufficient to activate pituitary stem cells

Conditional deletion of LATS1/2 kinases in the pituitary has revealed how these promote an expansion of SOX2+ stem cells in the embryonic and postnatal gland at the expense of differentiation. To establish if this effect was mediated through YAP alone, we used the tetracycline-controlled conditional YAP-TetO system to promote YAP (S127A) protein levels in postnatal pituitaries of *Hesx1^{Cre/+}; R26^{rtTA/+};Col1a1^{tetO-Yap/+}* mice. We treated YAP-TetO animals with doxycycline from P21 to P105

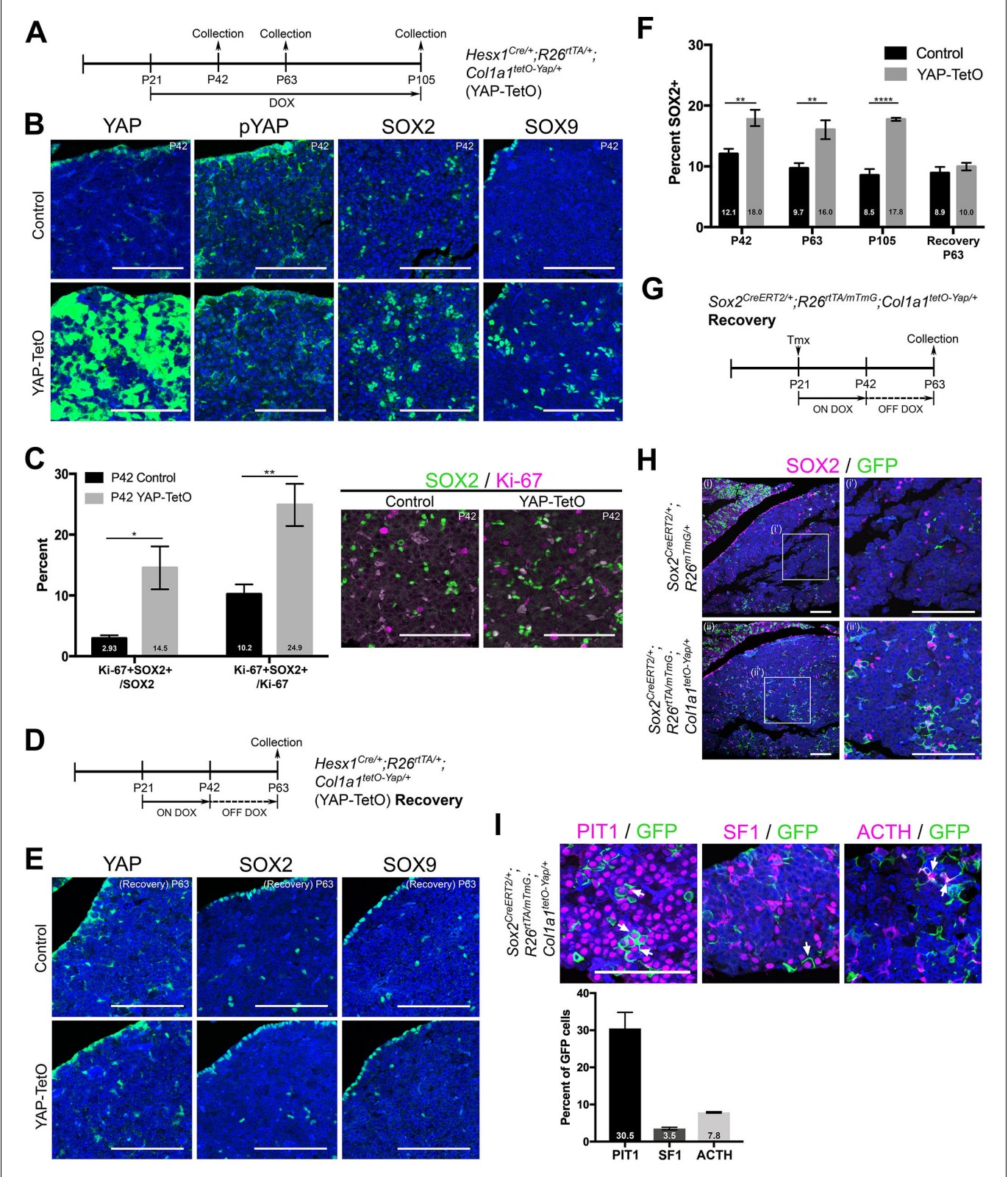

**Figure 5.** Postnatal expression of constitutively active YAP increases leads to an activation of SOX2+ pituitary stem cells. (**A**) Schematic outlining the time course of doxycycline (DOX) treatment administered to *Hesx1^Cre/+^;R26^rtTA/+^;Col1a1^tetO-Yap/+^* (YAP-TetO) and *Hesx1^+/+^;R26^rtTA/+^;Col1a1^tetO-Yap/+^* controls to drive expression of YAP-S127A in mutant pituitaries. (**B**) At P42 (3 weeks of treatment), immunofluorescence staining on frontal anterior pituitary sections detects strong total YAP expression in YAP-TetO mutants compared to the control and no increase in pYAP(S127).
*Figure 5 continued on next page*

Figure 5 continued

Immunofluorescence for SOX2 and SOX9 reveals an expanded population of stem cells in YAP-TetO compared to control (quantification in F). (C) Graph showing the percentage of double Ki-67+;SOX2+ cells as a proportion of the total SOX2+ (p=0.027 (*)) or Ki-67+ (p=0.006 (**)) populations at P42 (n = 3 pituitaries per genotype). There is an increase in the numbers of cycling SOX2 cells in YAP-TetO mutant compared to controls. The image shows a representative example of double immunofluorescence staining against Ki-67 and SOX2 in a control and YAP-TetO section. (D) Schematic outlining the time course of doxycycline (DOX) treatment administered to $Hesx1^{Cre/+}$;$R26^{rtTA/+}$;$Col1a1^{tetO-Yap/+}$ (YAP-TetO) and $Hesx1^{+/+}$;$R26^{rtTA/+}$; $Col1a1^{tetO-Yap/+}$ controls to drive expression of YAP-S127A in mutant pituitaries for three weeks, followed by a three-week recovery period in the absence of DOX. (E) Immunofluorescence staining against YAP, SOX2 and SOX9 on control and YAP-TetO pituitaries treated as in D, shows comparable expression of YAP, SOX2 and SOX9 between genotypes. (F) Graph of quantification of SOX2+ cells as a percentage of total nuclei in control and YAP-TetO pituitaries at P42 p=0.0014 (**); P63 p=0.0044 (**); P105 p<0.0001(****) (n = 3 pituitaries per genotype). Following the Recovery treatment scheme in D, there is no significant difference in the numbers of SOX2+ cells between genotypes. (G) Schematic outlining the time course of tamoxifen induction and doxycycline (DOX) treatment administered to $Sox2^{CreERT2/+}$;$R26^{rtTA/mTmG}$;$Col1a1^{tetO-Yap/+}$ (mutant) and $Sox2^{CreERT2/+}$; $R26^{mTmG/+}$;$Col1a1^{+/+}$ (control) animals to drive expression of YAP-S127A in SOX2+ cells of mutants. (H) Lineage tracing of SOX2+ cells and immunofluorescence staining against SOX2 and GFP shows an expansion of GFP+ cells compared to controls at P63, where a proportion of cells are double-labelled. (I) Immunofluorescence staining against commitment markers PIT1, SF1 and terminal differentiation marker ACTH (TPIT lineage) together with antibodies against GFP detects double-labelled cells (arrows) across all three lineages in $Sox2^{CreERT2/+}$;$R26^{rtTA/mTmG}$;$Col1a1^{tetO-Yap/+}$ pituitaries following the recovery period. Graph of quantification of GFP+;PIT1+, GFP+;SF1+ and GFP+;ACTH+ cells as a percentage of total GFP + cells in $Sox2^{CreERT2/+}$;$R26^{rtTA/mTmG}$;$Col1a1^{tetO-Yap/+}$ pituitaries at P63. Scale bars 100 μm. Data in C. and F. represented as mean ±SEM, analysed with Two-Way ANOVA with Sidak's multiple comparisons. See also *Figure 5—figure supplement 1*.

DOI: https://doi.org/10.7554/eLife.43996.013

The following figure supplement is available for figure 5:

**Figure supplement 1.** Postnatal expression of constitutively active YAP leads to an activation of SOX2+ pituitary stem cells.

DOI: https://doi.org/10.7554/eLife.43996.014

(12 week treatment, *Figure 5A*). We did not observe the formation of tumours at any stage analysed (n = 12, *Figure 5—figure supplement 1A*). Similarly, we did not observe the formation of lesions when treating from P5. This is in contrast with the unequivocal tumour formation observed in *Sox2-CreERT2/+*;*Lats1fl/fl*;*Lats2fl/+* mice. Elevation of YAP protein levels was confirmed following three weeks of doxycycline treatment (P42), displaying patchy accumulation, likely a result of genetic recombination efficiencies (*Figure 5B*). Consistent with pathway activation, there was robust elevation in the expression of transcriptional targets *Cyr61* and *Ctgf* following treatment (*Figure 5—figure supplement 1B*), however at significantly lower levels compared to *Sox2CreERT2/+*;*Lats1fl/fl*;*Lats2fl/+* deletions (*Figure 5—figure supplement 1E*), and there was no elevation in phosphorylated inactive YAP (*Figure 5B*).

Immunofluorescence against SOX2 demonstrated a significant increase in the number of SOX2 + cells as a proportion of the anterior pituitary (*Figure 5B,F*; 18.0% compared to 12.1% in controls, p=0.0014), a finding recapitulated by SOX9 that marks a subset of the SOX2 population (*Figure 5B*). This increase in the percentage of SOX2+ cells was maintained at all stages analysed (*Figure 5F*) and did not affect the overall morphology of the pituitary. At P42 we observed a significant increase in proliferation among the SOX2+ pituitary stem cells from 3% in controls to 15% in mutants (p=0.027). SOX2+ cells make up 10% of all cycling cells (Ki-67%) in normal pituitaries, however in mutants this increased to 25%, suggesting a preferential expansion of the SOX2+ population, rather than an overall increase in proliferation (*Figure 5C*). No additional marked differences were observed in samples analysed at P63 (6 weeks of treatment, n = 3), however longer treatment (P21 to P105) resulted in sporadic regions of expanded SOX2+ cells (*Figure 5—figure supplement 1C*). These regions did not express the commitment marker PIT1 and were identifiable by haematoxylin/eosin staining. In contrast to tumour lesions generated following loss of LATS kinases, these were not proliferative, were positive for pYAP and did not accumulate high levels of YAP/TAZ (n = 6 lesions). Together these results suggest that the sustained expression of constitutive active YAP can activate the proliferation of SOX2 stem cells, but in contrast to deletion of LATS1, this alone is not oncogenic.

To establish if the expansion of pituitary stem cells following forced expression of YAP is reversible, we administered doxycycline to YAP-TetO animals for three weeks (P21 to P42) by which point there is a robust response, followed by doxycycline withdrawal for three weeks (until P63) to allow sufficient time for YAP levels to return to normal (scheme *Figure 5D*). Immunofluorescence against total YAP protein confirmed restoration of the normal YAP expression pattern and levels after

recovery (*Figure 5E*), and mRNA in situ hybridisation detected a reduction in expression of YAP/TAZ targets *Cyr61* and *Ctgf* (*Figure 5—figure supplement 1D*). Following recovery from high levels of YAP, the number of SOX2+ cells reduced to comparable levels as in controls (around 10% of the total anterior pituitary) (*Figure 5E,F*). This suggests that the effects of YAP overexpression on the stem cell population are transient following three weeks of treatment (*Figure 5F*).

Finally, to determine if SOX2+ cells could differentiate into hormone-producing cells after the reduction in YAP levels, we expressed constitutive active YAP only in SOX2+ cells while lineage tracing this population (*Sox2$^{CreERT2/+}$;R26$^{rtTA/mTmG}$;Col1a1$^{tetO-Yap/+}$*). We induced SOX2+ cells by low-dose tamoxifen administration at P21 and treated with doxycycline for three weeks, followed by doxycycline withdrawal for a further three weeks (*Figure 5G*). Larger clones of SOX2 derivatives were observed at P63 in *Sox2$^{CreERT2/+}$;R26$^{rtTA/mTmG}$;Col1a1$^{tetO-Yap/+}$* animals compared to controls, and these still contained SOX2+ cells (*Figure 5H*). Following withdrawal, we were able to detect GFP+ derivatives of SOX2+ cells, which had differentiated into the three lineages (PIT1, SF1 and ACTH, marking corticotrophs of the TPIT lineage) (*Figure 5I*). Taken together, these findings confirm that sustained expression of YAP is sufficient to maintain the SOX2+ state and promote activation of normal SOX2+ pituitary stem cells in vivo, driving expansion of this population.

## Discussion

Here we establish that regulation of LATS/YAP/TAZ signalling is essential during anterior pituitary development and can influence the activity of the stem/progenitor cell pool. LATS kinases, mediated by YAP and TAZ, are responsible for controlling organ growth, promoting an undifferentiated state and repressing lineage commitment. Loss of both *Lats1* and *Lats2*, encoding potent tumour suppressors, leads to dramatic tissue overgrowth during gestation, revealing a function for these enzymes in restricting growth during pituitary development. The involvement of YAP/TAZ and dysfunction of the kinase cascade is emerging in multiple paediatric cancers, which are often developmental disorders (*Ahmed et al., 2017*).

Loss of *LATS1* heterozygosity has been reported in a range of human tumours (*Lee et al., 1990*; *Chen et al., 2005*; *Theile et al., 1996*; *Mazurenko et al., 1999*) leading to an increase in YAP/TAZ protein levels. Previous global deletion of *Lats1* in mice resulted in a variety of soft tissue sarcomas and stromal cell tumours (*St John et al., 1999*). The anterior lobe of these animals appeared hyperplastic with poor endocrine cell differentiation leading to combined hormone deficiencies, but the presence of tumours was not noted. We report that loss of *Lats1* alone is sufficient to drive anterior and intermediate lobe tumour formation. This phenotype is accelerated following additional deletion of one copy of *Lats2*. Phenotypically identical tumour lesions were generated when the genetic deletions were carried out embryonically in RP, or at postnatal stages. Interestingly, tissue-specific loss of *Stk3* and *Stk4*, which regulate LATS activation in other tissues (*Hu et al., 2017*), did not lead to any pituitary defects despite reduction in STK3/4 levels. These data suggest that perhaps the residual activity of STK3/4 is sufficient for LATS1/2 activation. Alternatively, regulation of LATS1/2 by kinases other than STK3/4 is possible in the pituitary, meaning deletion of *Stk3/4* alone is insufficient to result in significant LATS function impairment. Similar situations have been reported in other organs where LATS are functioning (*Hu et al., 2017*). The resulting non-secreting tumours in our mouse models are composed predominantly of SOX2+ stem cells and display signs of squamous differentiation. Rare cases of squamous cell carcinoma have been reported as primary pituitary tumours (*Saeger et al., 2007*), but more frequently, arising within cysts that are normally non-neoplastic epithelial malformations (*Lewis et al., 1983*; *O'Neill et al., 2016*). In the embryonic YAP-TetO model, where constitutive active YAP (S127A) was expressed during pituitary development, cysts phenocopying Rathke's cleft cyst, develop by postnatal stages. Target elevation is not as high in YAP-TetO pituitaries, as following the deletion of LATS1/2, indicating that signalling levels are likely to be critical for progression between these phenotypes.

Although human pituitary carcinomas are only diagnosed as such after metastasis, the tumours generated in our LATS1/2 mouse models fit their histopathological profile. Genetic lineage tracing identified SOX2+ cells as the cell of origin of the tumours; this observation could have ramifications regarding involvement of the LATS/YAP/TAZ pathway in the establishment or progression of human pituitary tumours composed of uncommitted cells. In cancer stem cells of osteosarcoma and glioblastoma, SOX2 antagonises upstream Hippo activators, leading to enhanced YAP function (*Basu-*

*Roy et al., 2015*). We recently reported enhanced expression of YAP/TAZ in a range of non-functioning human pituitary tumours, compared to functioning adenomas, and that *Lats1* knock-down in GH3 pituitary mammosomatotropinoma cells results in repression of the *Gh* and *Prl* promoters (*Xekouki et al., 2019*). Therefore, YAP/TAZ, perhaps in a positive feedback loop with SOX2, are likely to function both to promote the maintenance of an active pituitary stem cell state as well as to inhibit differentiation.

By dissecting the downstream requirement for YAP in pituitary regulation by the LATS/YAP/TAZ axis, we found that expression of constitutively active YAP (S127A) is sufficient to push SOX2+ pituitary stem cells into an activated state, leading to expansion of the stem cell cohort (see Model, *Figure 6*). YAP has previously been indicated to promote the stem cell state in other tissues, for example pancreas, neurons and mammary glands (*Panciera et al., 2016*). However, this does not fully recapitulate the LATS deletion phenotypes, as it did not lead to the formation of tumours

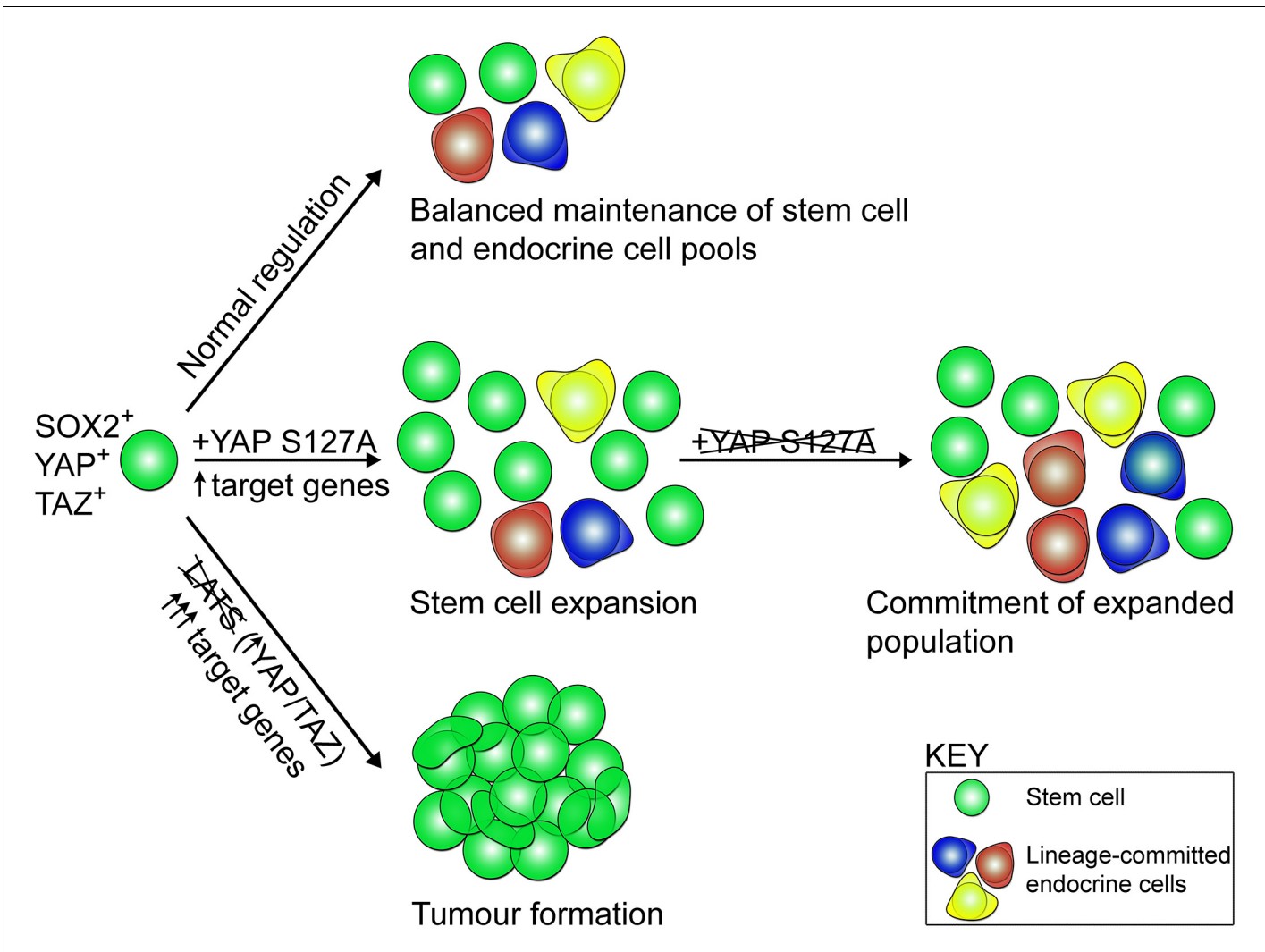

**Figure 6.** Model of stem cell activity following regulation by the LATS/YAP/TAZ cascade in the anterior pituitary. SOX2+ pituitary stem cells express YAP and TAZ (green spheres). During normal developmental and postnatal expansion (normal regulation), pituitary stem cells are maintained as a balanced pool while generating endocrine cells of three committed lineages (red, blue, yellow). Expression of constitutively active YAP-S127A in pituitary stem cells leads to an elevation in target gene expression, an expansion of pituitary stem cell numbers and maintenance of the SOX2+ state, preventing lineage commitment. When YAP-S127A expression ceases, commitment into the endocrine lineages takes place. Genetic deletion of LATS kinases (LATS1 as well as one or two copies of LATS2), results in YAP and TAZ accumulation, major elevation in target gene expression, repression of lineage commitment, continued expansion of SOX2+ cells and tumour formation.
DOI: https://doi.org/10.7554/eLife.43996.015

during the time course of YAP activation (12 weeks). Interestingly, since the levels of target activation are significantly greater in *Lats1/2* deletions that in YAP-TetO activation, initiation of tumourigenesis may be associated with levels of signalling rising above a threshold. However, the temporal control of expressing the mutation is critical, as seen in other tumour models (*Han et al., 2016*). Instead, the findings identify an isolated role for YAP in promoting the expansion of the SOX2 + stem cell pool and restoring their proliferative potential to levels akin to the most active state during postnatal pituitary growth. Activity of YAP/TAZ is reduced in dense tissues, resulting in a decrease in stemness. One mechanism through which this is achieved is by crosstalk with other signalling pathways regulating stem cell fate (*Papaspyropoulos et al., 2018*; *Heallen et al., 2011*). For example, a decrease in YAP/TAZ activity removes inhibition on Notch signalling, resulting in higher levels of differentiation and a drop in stem cell potential (*Totaro et al., 2017*). In the pituitary, Notch plays a role in the maintenance of the SOX2 stem cell compartment and is involved in regulating differentiation (*Zhu et al., 2015*; *Nantie et al., 2014*; *Cheung et al., 2013*; *Batchuluun et al., 2017*). The downstream mechanisms of YAP action on SOX2+ pituitary stem cells, as well as the likely crosstalk with other signalling pathways remain to be explored.

In summary, our findings highlight roles for LATS/YAP/TAZ in the regulation of pituitary stem cells, where fine-tuning of their expression can make the difference between physiological stem cell re-activation and tumourigenesis, of relevance to other organs. We reveal this axis is involved in the control of cell fate commitment, regulation of regenerative potential and promotion of tumourigenesis. These findings can aid in the design of treatments against pituitary tumours and in regenerative medicine approaches targeting the regulation of endogenous stem cells.

# Materials and methods

## Key resources table

| Reagent type (species) or resource | Designation | Source or reference | Identifiers | Additional information |
|---|---|---|---|---|
| Genetic reagent (*M. musculus*) | Hesx1$^{Cre/+}$ | *Andoniadou et al., 2007* | RRID:MGI:5314529 | |
| Genetic reagent (*M. musculus*) | Sox2$^{CreERT2/+}$ | *Andoniadou et al., 2013* | MGI:5512893 | |
| Genetic reagent (*M. musculus*) | Lats1$^{fl/fl}$ | Jackson Laboratory | Stock #: 024941, RRID: MGI:5568576 | |
| Genetic reagent (*M. musculus*) | Lats2$^{fl/fl}$ | Jackson Laboratory | Stock #: 025428, RRID: MGI:5568577 | |
| Genetic reagent (*M. musculus*) | Stk4$^{fl/fl}$;Stk3$^{fl/fl}$ | Jackson Laboratory | Stock #: 017635, RRID: MGI:5301573 | *Lu et al., 2010* |
| Genetic reagent (*M. musculus*) | R26$^{rtTA/+}$ | Jackson Laboratory | Stock: #: 016999 RRID: MGI:5292520 | *Yu et al., 2005* |
| Genetic reagent (*M. musculus*) | Col1a1$^{tetO-Yap/+}$ | *Jansson and Larsson, 2012* | MGI:5430522 | |
| Genetic reagent (*M. musculus*) | R26$^{mTmG/+}$ | Jackson Laboratory | Stock #: 007576 RRID:MGI:3722405 | *Muzumdar et al., 2007* |
| Genetic reagent (*M. musculus*) | Taz$^{-/-}$ | Jackson Laboratory | Stock #: 011120, RRID: MGI:4420900 | *Tian et al., 2007* |
| Genetic reagent (*M. musculus*) | Yap$^{fl/fl}$ | *Schlegelmilch et al., 2011* | MGI:5316446 | |
| Antibody | Rabbit polyclonal anti-TAZ | Atlas Antibodies | Cat# HPA007415 RRID:AB_1080602 | IF: 1:1000 |
| Antibody | Rabbit polyclonal anti-YAP | Cell Signaling Technology | Cat# 4912S RRID:AB_2218911 | IF: 1:1000 |

*Continued on next page*

*Continued*

| Reagent type (species) or resource | Designation | Source or reference | Identifiers | Additional information |
|---|---|---|---|---|
| Antibody | Rabbit polyclonal anti-pYAP | Cell Signaling Technology | Cat# 4911S RRID:AB_2218913 | IF: 1:1000 |
| Antibody | Rabbit polyclonal anti-SOX2 | Abcam | Cat# ab97959 RRID:AB_2341193 | IF: 1:2000 |
| Antibody | Rat monoclonal anti-EMCN (V.7C7.1) | Abcam | Cat# ab106100 RRID:AB_10859306 | IF: 1:1000 |
| Antibody | Chicken polyclonal anti-GFP | Abcam | Cat# ab13970 RRID:AB_300798 | IF: 1:300 |
| Antibody | Goat polyclonal anti-SOX2 | Immune Systems Limited | Cat# GT15098 RRID:AB_2732043 | IF: 1:250 |
| Antibody | Rabbit monoclonal anti-Ki-67 | Abcam | Cat# ab16667 RRID:AB_302459 | IF: 1:300 |
| Antibody | Rabbit polyclonal anti-ARL13B | Proteintech Group | Cat# 17711–1-AP, RRID:AB_2060867 | IF: 1:100 |
| Antibody | Mouse monoclonal anti-Acetylated-αTUB | Sigma-Aldrich | Cat# MABT868 | IF: 1:200 |
| Antibody | Rabbit monoclonal anti-SOX9 | Abcam | Cat# ab185230 RRID:AB_2715497 | IF: 1:300 |
| Antibody | Rabbit monoclonal anti-Active YAP EPR19812 | Abcam | Cat# ab205270 | IF: 1:300 |
| Antibody | Rabbit polyclonal anti-PIT1 | Prof. S Rhodes (Indiana University) | | IF: 1:1000 |
| Antibody | Rabbit polyclonal anti-TPIT | Prof. J Drouin (Montreal IRCM) | | IF: 1:1000 |
| Antibody | Mouse monoclonal anti-SF1 | Life Technologies (Thermo Fisher Scientific) | Cat# N1665 RRID:AB_2532209 | IF: 1:200 |
| Antibody | Rabbit polyclonal anti-gamma H2A.X (phospho S139) | Abcam | Cat# ab2893 RRID:AB_303388 | IF: 1:1000 |
| Antibody | Rabbit polyclonal anti-STK3/4 | Bethyl Laboratories | Cat# A300-466A RRID:AB_2148394 | WB: 1:5000 |
| Antibody | Mouse monoclonal anti-Cyclophilin B (Clone# 549205) | R and D Systems | Cat# MAB5410 RRID:AB_2169416 | WB: 1:1000 |
| Antibody | Rabbit monoclonal anti-Vimentin (D21H3) | Cell Signaling Technology | Cat# 5741 RRID:AB_10695459 | IF: 1:300 |
| Antibody | Biotinylated Goat polyclonal anti-rabbit | Abcam | Cat# ab6720 RRID:AB_954902 | IF: 1:350 |
| Antibody | Goat polyclonal anti-chicken Alexa Fluor 488 | Life Technologies (Thermo Fisher Scientific) | Cat# A11039 RRID:AB_2534096 | IF: 1:300 |

*Continued on next page*

*Continued*

| Reagent type (species) or resource | Designation | Source or reference | Identifiers | Additional information |
|---|---|---|---|---|
| Antibody | Goat polyclonal anti-rat Alexa Fluor 555 | Life Technologies (Thermo Fisher Scientific) | Cat# A21434 RRID:AB_2535855 | IF: 1:300 |
| Antibody | Biotinylated Goat polyclonal anti-mouse | Abcam | Cat# ab6788 RRID:AB_954885 | IF: 1:350 |
| Antibody | Donkey polyclonal anti-goat Alexa Fluor 488 | Abcam | Cat# ab150133 | IF: 1:300 |
| Antibody | Streptavidin Alexa Fluor 555 | Life Technologies | Cat# S21381 RRID:AB_2307336 | IF: 1:500 |
| Antibody | Goat HRP-linked anti-rabbit | Cell Signaling Technology | Cat# 7074 RRID:AB_2099233 | WB: 1:2000 |
| Antibody | Goat HRP-linked anti-mouse | Cell Signaling Technology | Cat# 7076 RRID:AB_330924 | WB: 1:2000 |
| Antibody | Mouse monoclonal anti-AE1/AE3 | Dako | Cat# M351529 | IHC: 1:100 |
| Antibody | Mouse monoclonal anti-Chromogranin | Dako | Cat# M086901 | IHC: 1:400 |
| Antibody | Mouse monoclonal anti-NCAM | Novocastra | Cat# NCL-L-CD56-504 | IHC 1:15 |
| Antibody | Mouse monoclonal anti-NSE | Dako | Cat# M087329 | IHC 1:1000 |
| Antibody | Mouse monoclonal anti-p63 | A Menarini Diagnostics | Cat# MP163 | IHC 1:100 |
| Antibody | Mouse monoclonal anti-Synaptophysin | Dako | Cat# M731529 RRID:AB_2687942 | IHC 1:2 |
| Commercial assay or kit | TSA kit | Perkin Elmer | Cat# NEL753001KT | |
| Commercial assay or kit | TSA Blocking Reagent | Perkin Elmer | Cat# FP1020 | |
| Commercial assay or kit | ABC kit | Vector Laboratories | Cat# Vector PK-6100 RRID:AB_2336819 | |
| Commercial assay or kit | BCA assay | Thermo Fisher | Cat# 23227 | |
| Commercial assay or kit | UltraView Universal DAB Detection Kit | Ventana Medical Systems | Cat# 760–500 | |
| Commercial assay or kit | VectaFluor Excel R.T.U. Antibody Kit, DyLight 488 Anti-Mouse | Vector Laboratories | Cat# DK-2488 RRID:AB_2336775 | |
| Chemical compound, drug | Doxycycline hyclate | Alfa Aesar | Cat# J60579 | 2 mg/ml |
| Chemical compound, drug | Sucrose | Sigma-Aldrich | Cat# S0389 | 10 mg/ml |
| Chemical compound, drug | Tamoxifen | Sigma-Aldrich | Cat# T5648 | 0.15 mg/g |
| Chemical compound, drug | Hoechst 33342 | Life Technologies | Cat# H3570 | 1:10000 |

*Continued on next page*

*Continued*

| Reagent type (species) or resource | Designation | Source or reference | Identifiers | Additional information |
|---|---|---|---|---|
| Chemical compound, drug | Laemmli buffer | Bio-Rad | Cat# 1704156 | |
| Chemical compound, drug | Clarity Western ECL Substrate | Bio-Rad | Cat# 170–5060 | |
| Chemical compound, drug | Alcian Blue | Alfa Aeser | Cat# J60122 | 1% |
| Chemical compound, drug | Acetic acid | VWR | Cat# 20103 | 3% |
| Chemical compound, drug | Periodic acid | VWR | Cat# 29460 | 1% |
| Chemical compound, drug | Schiff's reagent | Thermo Fisher Scientific | Cat# 88017 | |
| Software, algorithm | GraphPad Prism | GraphPad Software (www.graphpad.com) | RRID:SCR_015807 | |
| Software, algorithm | Fiji | *Schindelin et al., 2012* (Fiji.sc) | RRID:SCR_002285 | |
| Software, algorithm | ImageLab | BioRad | | |
| Other | Probe: *Ctgf* | ACDBio | Cat# 314541 | |
| Other | Probe: *Cyr61* | ACDBio | Cat# 429001 | |
| Other | Probe: *Lats2* | ACDBio | Cat# 420271 | |
| Other | Probe: *Nr5a1* | ACDBio | Cat# 445731 | |
| Other | Probe: *Tbx19* | ACDBio | Cat# 484741 | |
| Other | Probe: *Pou1f1* | ACDBio | Cat# 486441 | |

## Animals

Animal husbandry was carried out under compliance of the Animals (Scientific Procedures) Act 1986, Home Office license and KCL ethical review approval.

The $Hesx1^{Cre/+}$ *Andoniadou et al., 2007*, $Sox2^{CreERT2/+}$ *Andoniadou et al., 2013*, $Yap^{fl/fl}$ [25], $Taz^{-/-}$ *Tian et al., 2007* (JAX:011120), $R26^{mTmG/+}$ *Muzumdar et al., 2007* (JAX:007576), $ROSA26^{rtTA/+}$ *Yu et al., 2005* (JAX:016999), $Col1a1^{tetO-Yap/+}$ *Jansson and Larsson, 2012* (MGI:5430522), $Stk3^{fl/fl}$; $Stk4^{fl/fl}$ *Lu et al., 2010* (JAX:017635), and $Lats1^{fl/fl}$ *Heallen et al., 2011* (JAX:024941) and $Lats2^{fl/fl}$ *Heallen et al., 2011* (JAX:025428) have been previously described.

Tamoxifen (Sigma, T5648) was administered to experimental mice by intraperitoneal injection at a single dose of 0.15 mg/g body weight, or two equal doses on sequential days, depending on the experiment. Mice for growth studies were weighed every week. For embryonic studies, timed matings were set up where noon of the day of vaginal plug was designated as 0.5dpc.

For YAP-TetO experiments, crosses between $Hesx1^{Cre/+}$;$R26^{+/+}$;$Col1a1^{+/+}$ and $Hesx1^{+/+}$;$R26^{rtTA/rtTA}$;$Col1a1^{tetO-Yap/\ tetO-Yap}$ animals were set up to generate $Hesx1^{Cre/+}$;$R26^{rtTA/+}$;$Col1a1^{tetO-Yap/+}$ offspring (hereby YAP-TetO) and control littermates, or crosses between $Sox2^{CreERT2/+}$;$R26^{mTmG/mTmG}$;$Col1a1^{+/+}$ and $Sox2^{+/+}$; $R26^{rtTA/rtTA}$;$Col1a1^{tetO-Yap/\ tetO-Yap}$ animals were set up to generate $Sox2^{CreERT2/+}$;$R26^{rtTA/mTmG}$;$Col1a1^{tetO-Yap/+}$ offspring. While treated with the tetracycline analogue doxycycline, YAP-TetO expressed rtTA from the *ROSA26* locus in *Cre*-derived cells, enabling YAP-S127A expression from the *Col1a1* locus. For embryonic studies between 5.5dpc and 15.5dpc (scheme, *Figure 1A*), doxycycline (Alfa Aesar, J60579) was administered to pregnant dams in the drinking water at 2 mg/ml, supplemented with 10% sucrose. For postnatal analyses animals were treated with

doxycycline or vehicle (DMSO) as described, from the ages specified for individual experiments on the *Hesx1^{Cre/+}* driver, or directly following tamoxifen administration for animals on the *Sox2^{CreERT2/+}* driver. Both male and female mice and embryos where included in the studies.

## Tissue preparation

Embryos and adult pituitaries were fixed in 10% neutral buffered formalin (Sigma) overnight at room temperature. The next day, tissue was washed then dehydrated through graded ethanol series and paraffin-embedded. Embryos up to 13.5dpc were sectioned sagittal and all older embryo and post-natal samples were sectioned frontal, at a thickness of 7 μm for immunofluorescence staining, or 4 μm for RNAscope mRNA in situ hybridisation.

## RNAscope mRNA in situ hybridisation

Sections were selected for the appropriate axial level, to include Rathke's pouch or pituitary, as described previously (*Lodge et al., 2016*). The RNAscope 2.5 HD Reagent Kit-RED assay (Advanced Cell Diagnostics) was used with specific probes: *Ctgf, Cyr61, Lats2* (all ACDBio).

## H and E staining

Sections were dewaxed in histoclear and rehydrated through graded ethanol series from 100% to 25% ethanol, then washed in distilled $H_2O$. Sections were stained with Haematoxylin QS (Vector #H3404) for 1 min, and then washed in water. Slides were then stained in eosin in 70% ethanol for 2 min and washed in water. Slides were dried and coverslips were mounted with VectaMount permanent mounting medium (Vector Laboratories H5000).

## Immunofluorescence and immunohistochemistry

Slides were deparaffinised in histoclear and rehydrated through a descending graded ethanol series. Antigen retrieval was performed in citrate retrieval buffer pH6.0, using a Decloaking Chamber NXGEN (Menarini Diagnostics) at 110°C for 3mins. Tyramide Signal Amplification (TSA) was used for staining using antibodies against YAP (1:1000, Cell Signaling #4912S), pYAP (1:1000, Cell Signaling #4911S), TAZ (1:1000, Atlas Antibodies #HPA007415) and SOX2 (1:2000, Abcam ab97959) with EMCN (1:1000, Abcam ab106100) staining as follows: sections were blocked in TNB (0.1M Tris-HCl, pH7.5, 0.15M NaCl, 0.5% Blocking Reagent (Perkin Elmer FP1020)) for 1 hr at room temperature, followed by incubation with primary antibody at 4°C overnight, made up in TNB. Slides were washed three times in TNT (0.1MTris-HCl pH7.5, 0.15M NaCl, 0.05% Tween-20) then incubated with secondary antibodies (biotinylated anti-rabbit (1:350 Abcam ab6720) and anti-Rat Alexa Fluor 555 (1:300, Life Technologies A21434) for 1 hr at room temperature and Hoechst (1:10000, Life Technologies H3570). Slides were washed again then incubated in ABC reagent (ABC kit, Vector Laboratories PK-6100) for 30 mins, followed by incubation with TSA conjugated fluorophore (Perkin Elmer NEL753001KT) for ten minutes. Slides were washed and mounted with VectaMount (Vector Laboratories H1000).

For regular immunofluorescence, sections were blocked in blocking buffer (0.15% glycine, 2 mg/ml BSA, 0.1% Triton-X in PBS), with 10% sheep serum (donkey serum for goat SOX2 antibody) for 1 hr at room temperature, followed by incubation with primary antibody at 4°C overnight, made up in blocking buffer with 1% serum. Primary antibodies used were against SOX2 (1:250, Immune Systems Ltd GT15098), active YAP (1:300, Abcam ab205270), GFP (1:300, Abcam ab13970), Ki-67 (1:300, Abcam ab16667), SOX9 (1:300, Abcam ab185230), PIT1 (1:1000, Gift from S. Rhodes, Indiana University), TPIT (1:1000, Gift from J. Drouin, Montreal), SF1 (1:200, Life Technologies N1665), Gamma H2A.X (1:1000, Abcam ab2893), Vimentin (1:300, Cell Signaling #5741), Caspase (1:300, Cell Signaling #9661S). Slides were washed in PBST then incubated with secondary antibodies for 1 hr at room temperature. Appropriate secondary antibodies were incubated in blocking buffer for 1 hr at room temperature (biotinylated anti-rabbit (1:350, Abcam ab6720), biotinylated anti-mouse (1:350, Abcam ab6788), anti-chicken 488 (1:300, Life Technologies A11039), anti-goat 488 (1:300, Abcam ab150133). Slides were washed again using PBST and incubated with fluorophore-conjugated Streptavidin (1:500, Life Technologies S21381 or S11223) for 1 hr at room temperature, together with Hoechst (1:10000, Life Technologies H3570). Slides were washed in PBST and mounted with Vecta-Mount (Vector Laboratories, H1000).

Immunohistochemistry for the remaining antigens were undertaken on a Ventana Benchmark Autostainer (Ventana Medical Systems) using the following primary antibodies and antigen retrieval: AE1/AE3 (1:100, Dako M351529), CC1 (36 min, Ventana Medical Systems 950–124); Chromogranin (1:400, Dako M086901), CC1 (36 min, Ventana Medical Systems 950–124); NCAM (1:15, Novocastra NCL-L-CD56-504), CC1 (64 min, Ventana Medical Systems 950–124); NSE (1:1000, Dako M087329), CC1 (36 min, Ventana Medical Systems 950–124); p63 (1:100, A. Menarini Diagnostics), CC1 (64 min, Ventana Medical Systems 950–124) and Synaptophysin (1:2, Dako M731529), CC2 (92 min, Ventana Medical Systems 950–124). Targets were detected and viewed using the ultraView Universal DAB Detection Kit (Ventana Medical Systems, 760–500) according to manufacturer's instructions.

## Alcian blue with periodic Acid-Schiff staining (AB/PAS)

Following deparaffinisation and rehydration, sections were taken through distilled water then placed in Alcian Blue solution (1% Alcian Blue (Alfa Aeser J60122) in 3% acetic acid (VWR International 20103)) for 20 min. Sections were then placed in 1% periodic acid (VWR 29460) for 10 min, washed in distilled water and transferred to Schiff's reagent (Thermo Fisher Scientific 88017) for 10 min, followed by washing in distilled water for 5 min. Sections were then routinely dried, cleared and mounted.

## Western blotting

Dissected anterior pituitaries were flash frozen in liquid nitrogen and stored at $-80°C$. Frozen pituitaries were each lysed in 30 µl of lysis buffer (5 mM Tris, 150 mM NaCl, 1% protease and phosphatase inhibitor (Abcam ab201119), 5 µM EDTA, 0.1% Triton-X, pH7.6) and sonicated at 40% power, twice for ten cycles of: two seconds on/two seconds off, using a Vibra-Cell Processor (Sonics). Protein concentration was determined using the Pierce BCA protein assay kit (Thermo #23227) and all samples were diluted to 4 mg/ml in Laemmli buffer (Biorad #161–0747). Proteins were denatured at 95°C for 5 min. Samples were run on a 10% Mini-PROTEAN TGX polyacrylamide gel (BioRad #4561033), then transferred using Trans-Blot Turbo transfer machine (BioRad) onto polyvinylidene difluoride membranes (BioRad #1704156). Membranes were blocked with 5% non-fat dairy milk (NFDM) in TBST (20 mM Tris, 150 mM NaCl, 0.1% Tween-20, pH7.6), cut, then incubated with primary antibodies overnight at 4°C as follows: anti-STK3/STK4 (1:5000, Bethyl Laboratories #A300-466A) or Cyclophilin B (1:1000, R and D Systems #MAB5410) in 5%NFDM. The next day, membranes were washed in TBST, incubated with secondary antibodies HRP-conjugated anti-Rabbit (1:2000, Cell Signaling #7074) or HRP-conjugated anti-Mouse (1:2000, Cell Signaling #7076) in 5% NFDM for 1 hr at room temperature. After washing in TBST, membranes were treated with Clarity Western ECL substrate (Biorad #170–5060) and bands visualised using the ChemiDoc Touch Imaging System (BioRad). Protein abundance was analysed using ImageLabs (BioRad).

## Imaging

Wholemount images were taken with a MZ10 F Stereomicroscope (Leica Microsystems), using a DFC3000 G camera (Leica Microsystems). For bright field images, stained slides were scanned with Nanozoomer-XR Digital slide scanner (Hamamatsu) and images processed using Nanozoomer Digital Pathology View. Fluorescent staining was imaged with a TCS SP5 confocal microscope (Leica Microsystems) and images processed using Fiji (*Schindelin et al., 2012*).

## Quantifications and statistics

Cell counts were performed manually using Fiji cell counter plug-in; 5–10 fields were counted per sample, totalling over 1500 nuclei, across 3–7 pituitaries. Statistical analyses and graphs were generated in GraphPad Prism (GraphPad Software) and the following tests were performed to determine significance: Student's *t*-tests between controls and mutants for *Figures 1D* and *2D*, *Figure 2—figure supplement 2D and 2E* (n = 3 of each genotype), *Figure 4—figure supplement 1* (n = 4 of each genotype) and *Figure 5C* (n = 4–5 of each genotype); unpaired *t*-test for *Figure 2—figure supplement 2A* (n = 3 per genotype) and *Figure 2—figure supplement 2F* (n = 6 sections across two samples per genotype); two-tailed *t*-test for *Figure 3C* (n = 3 controls, seven mutants); two-way ANOVA with Sidak's multiple-comparison test for *Figure 5F* (n = 4–5 of each genotype). For quantification of target expression by RNAscope mRNA in situ hybridisation (*Figure 5—figure*

*supplement 1*), the area of positive staining (red fluorescence) from 4 µm sections was determined from images using thresholding in Fiji, and quantified as a percentage of total pituitary area in the same image. For statistical testing, one-way ANOVAs with Tukey's multiple comparisons were performed (n = 4 mutants per genotype). Error bars in graphs show ±standard error of the mean, unless otherwise indicated. Quantification of STK3/4 by western blot was carried out on two control (*Stk3^{fl/fl}*;*Stk4^{fl/fl}*) and three mutant (*Hesx1^{Cre/+}*;*Stk3^{fl/fl}*; *Stk4^{fl/fl}*) samples. A Student's t-test was carried out on normalised band intensities. Chi-squared tests were used to determine significant deviations of observed from expected genotypes presented as tables in *Supplementary files 1* and *2*.

## Acknowledgements

This study has been supported by grant MR/L016729/1 from the MRC and a Lister Institute Research Prize to CLA, by the Deutsche Forschungsgemeinschaft (DFG) within the CRC/Transregio 205/1 as well as GRK 2251 to CLA and SRB. EJL was supported by the King's Bioscience Institute and the Guy's and St Thomas' Charity Prize PhD Programme in Biomedical and Translational Science. JPR was supported by a Dianna Trebble Endowment Fund Dental Institute Studentship. We thank Prof. Jacques Drouin and Prof. Simon Rhodes for TPIT and PIT1 antibodies respectively, and the National Hormone and Peptide Program (Harbor–University of California, Los Angeles Medical Center) for providing some of the hormone antibodies used in this study. We thank Prof. Juan Pedro Martinez-Barbera, Dr Rocio Sancho and Dr Marika Charalambous for discussions and critical reading of the manuscript. The authors declare no conflict of interest.

## Additional information

### Funding

| Funder | Grant reference number | Author |
|---|---|---|
| Guy's and St Thomas' Charity | Prize PhD Programme | Emily J Lodge |
| King's College London | Dianna Trebble Endowment Fund Dental Institute Studentship | John P Russell |
| Deutsche Forschungsgemeinschaft | GRK 2251 | Stefan R Bornstein Cynthia Lilian Andoniadou |
| Deutsche Forschungsgemeinschaft | CRC/Transregio 205/1 | Stefan R Bornstein Cynthia Lilian Andoniadou |
| Medical Research Council | MR/L016729/1 | Cynthia Lilian Andoniadou |
| Lister Institute of Preventive Medicine | Prize Fellowship 2016 | Cynthia Lilian Andoniadou |

The funders had no role in study design, data collection and interpretation, or the decision to submit the work for publication.

### Author contributions

Emily J Lodge, Formal analysis, Validation, Investigation, Methodology, Writing—review and editing; Alice Santambrogio, John P Russell, Validation, Investigation; Paraskevi Xekouki, Formal analysis, Investigation; Thomas S Jacques, Resources, Investigation; Randy L Johnson, Resources, Methodology, Generation of the Stk3-flox, Stk4-flox, Lats1-flox and Lats2-flox mouse strains; Selvam Thavaraj, Resources, Investigation, Writing—review and editing; Stefan R Bornstein, Supervision, Funding acquisition, Writing—review and editing; Cynthia Lilian Andoniadou, Conceptualization, Supervision, Funding acquisition, Methodology, Writing—original draft, Writing—review and editing

### Author ORCIDs

Emily J Lodge (iD) http://orcid.org/0000-0003-0932-8515
Cynthia Lilian Andoniadou (iD) http://orcid.org/0000-0003-4311-5855

## Ethics

Animal experimentation: This study was performed in accordance to UK Home Office Regulations and experimental procedures were approved by the King's College Ethical Review Process.

## Decision letter and Author response

Decision letter https://doi.org/10.7554/eLife.43996.020
Author response https://doi.org/10.7554/eLife.43996.021

# Additional files

## Supplementary files

• Supplementary file 1. Table showing expected and observed frequency of genotypes from $Hesx1\text{-}Cre^{/+};Yap^{fl/fl};Taz^{+/-}$ x $Yap^{fl/fl};Taz^{+/-}$ at embryonic 15.5dpc and postnatal day 0–2. Embryonic: p=0.3471, Chi-square test (two tailed). Postnatal: p=0.0003 (***), Chi-square test (two tailed).
DOI: https://doi.org/10.7554/eLife.43996.016

• Supplementary file 2. Table showing expected and observed frequency of genotypes from $Hesx1\text{-}Cre^{/+};Lats1^{fl/fl};Lats2^{fl/+}$ x $Lats1^{fl/fl};Lats2^{fl/fl}$ and $Hesx1^{Cre/+};Lats1^{fl/+};Lats2^{fl/+}$ x $Lats1^{fl/fl};Lats2^{fl/+}$ at embryonic 15.5dpc and postnatal day 0–2. Embryonic: p<0.0001 (****), Chi-square test (two tailed). Postnatal: p<0.0001 (****), Chi-square test (two tailed).
DOI: https://doi.org/10.7554/eLife.43996.017

• Transparent reporting form
DOI: https://doi.org/10.7554/eLife.43996.018

## Data availability

According to UK research councils' Common Principles on Data Policy, all data generated or analysed during this study are included in the manuscript and supporting files.

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
