## [Decision Letter]

Thank you for submitting your article "Homeostatic and tumourigenic activity of *Sox2Sox2*^+^ pituitary stem cells is controlled by the LATS/YAP/TAZ cascade" for consideration by *eLife*. Your article has been reviewed by three peer reviewers, and the evaluation has been overseen by a Reviewing Editor and Marianne Bronner as the Senior Editor. The following individuals involved in review of your submission have agreed to reveal their identity: Sally A Camper (Reviewer #1); Elaine Emmerson (Reviewer #2).

The reviewers have discussed the reviews with one another and the Reviewing Editor has drafted this decision to help you prepare a revised submission.

Summary:

This paper assesses the roles of key candidates of the hippo pathway (MST, LATS, YAP, and TAZ) in pituitary stem cells (marked by *SOX2*). elegant loss-of-function and gain-of-function in vivo transgenic models coupled with immunofluorescence and high-powered microscopy to demonstrate that deletion of LATS kinases and subsequent upregulation of YAP/TAZ leads to a loss of regulation of SOX2+stem cells and uncontrolled expansion, ultimately leading to the formation of pituitary tumours. This well-written and nicely presented manuscript thoroughly examines the role of the hippo pathway in pituitary development and tumorigenesis. This work will broadly appeal to multiple fields including cancer and stem cell biology and could have significance in understanding the etiology of pediatric pituitary tumors.

Essential revisions:

1) Subsection “LATS, but not MST, kinases are required for normal pituitary development and differentiation”, second paragraph. Since the elimination of *Mst1* and *Mst2* did not produce any phenotype, it is important to demonstrate that the expression of these genes was actually knocked out. The *Hesx1^cre^*strain is reliable and well-characterized, but there are examples of genes that are resistant to being knocked out for unknown reasons. Can you show by PCR or some other method that the genes were knocked out? It is worthwhile adding some discussion about Mst.

2) Quantification of much of the imaging data would strengthen the manuscript.

Could the authors graphically show the cell quantification data for PIT1, TPIT and SF1 in Figure 1, rather than stating it in the text, as this way the differences and significance would be much easier for the reader to appreciate? Similarly, for Figure 2B; Ki-67, SOX9, Figure 2D; PIT1, TPIT and SF1, Figure 4D; PIT1, TPIT and SF1, and Figure 5I; PIT1/GFP, SF1/GFP and ACTH/GFP.

Did the authors quantify the number of cells expressing SOX2 and active YAP in Figure 1—figure supplement 1D? From the images presented it appears that the number of cells expressing active YAP is reduced in the *Hesx1^Cre/+^; Yap^fl/fl^; Taz*^-/-^ mutants, so quantification would be advantageous.

The authors state an overgrowth of the RP in *Hesx1^Cre/+^; Lats1^fl/fl^; Lats2^fl/fl^* mutants, could they measure and quantify the RP area and present this graphically?

3) It is not totally convincing that the phenotypic difference of postnatal *Lats1/2* removal (Figure 4) and YAP activation (Figure 5) is due to their intrinsic oncogenic potentials as the authors claim in Figure 6. It is possible that the inability of YAP to drive tumor formation from the Sox2+ stem cells is simply due to the levels of YAP transcriptional output between these two models. The authors should quantitatively measure the expression of YAP/TAZ target genes, such as CTGF, *Cyr61* and ANKRD1.

4) Are the controls displayed in Figure 3—figure supplement 14F normal tissue from wild-type littermate controls? Would the authors expect them to express markers of adenoma (Synaptophysin, NSE, Chromogranin) so strongly?

5) It is not clear why the authors restrict the analysis to embryonic stage in *Hesx1^CreER^;R26^rtTA^;YAP^teto^* mice after Dox induction at e5.5. Are they embryonic lethal? If not, what are the postnatal phenotypes in these animals?

---

## [Author Response]

Essential revisions:1) Subsection “LATS, but not MST, kinases are required for normal pituitary development and differentiation”, second paragraph. Since the elimination of Mst1 and Mst2 did not produce any phenotype, it is important to demonstrate that the expression of these genes was actually knocked out. The Hesx1^cre^ strain is reliable and well-characterized, but there are examples of genes that are resistant to being knocked out for unknown reasons. Can you show by PCR or some other method that the genes were knocked out? It is worthwhile adding some discussion about Mst.

We agree that it is important to show deletion of *Stk3* and *Stk4 (Mst2 and Mst1)*. To this end we have carried out western blotting using an antibody recognising total STK3/4 proteins. This revealed over 75% overall decrease in STK3/4 protein in total anterior pituitary lysates from P35 *Hesx1^Cre/+^;Stk3^fl/fl^; Hesx1^Cre/+fl/fl^* mutants compared with controls. The data and quantification are incorporated in Figure 2—figure supplement 1 and presented in the subsection “LATS, but not STK, kinases are required for normal pituitary development and differentiation”.

We have expanded discussion about STK in the second paragraph of the Discussion.

2) Quantification of much of the imaging data would strengthen the manuscript.Could the authors graphically show the cell quantification data for PIT1, TPIT and SF1 in Figure 1, rather than stating it in the text, as this way the differences and significance would be much easier for the reader to appreciate? Similarly, for Figure 2B; Ki-67, SOX9, Figure 2D; PIT1, TPIT and SF1, Figure 4D; PIT1, TPIT and SF1, and Figure 5I; PIT1/GFP, SF1/GFP and ACTH/GFP.

We have quantified the data requested in Figures 1, 2, 4 and 5. These are now presented graphically in panels: Figure 1D (lineage markers), Figure 2D (lineage markers), Figure 5I (lineage/GFP), Figure 5—figure supplement 1A (lineage). Additional data, presented as new Figure 1—figure supplement 1B-G (embryonic YAP-TetO postnatal phenotype, see response to Point 5, below) and Figure 2—figure supplement 2F (Ki-67 and SOX9), have also been quantified and presented graphically.

Did the authors quantify the number of cells expressing SOX2 and active YAP in Figure 1—figure supplement 1D? From the images presented it appears that the number of cells expressing active YAP is reduced in the Hesx1^Cre/+^; Yap^fl/fl^; Taz^-/-^ mutants, so quantification would be advantageous.

We agree with the reviewers that the number of SOX2+ cells expressing active YAP is reduced in these mutants and are keen to emphasise this point. We have quantified the cells expressing SOX2 and active YAP and present these in Figure 1—figure supplement 2D.

The authors state an overgrowth of the RP in Hesx1^Cre/+^; Lats1^fl/fl^; Lats2^fl/fl^ mutants, could they measure and quantify the RP area and present this graphically?

As suggested, we have measured the area of the RP in these mutants and presented this graphically in Figure 2—figure supplement 2E.

3) It is not totally convincing that the phenotypic difference of postnatal Lats1/2 removal (Figure 4) and YAP activation (Figure 5) is due to their intrinsic oncogenic potentials as the authors claim in Figure 6. It is possible that the inability of YAP to drive tumor formation from the Sox2+ stem cells is simply due to the levels of YAP transcriptional output between these two models. The authors should quantitatively measure the expression of YAP/TAZ target genes, such as CTGF, Cyr61 and ANKRD1.

Unfortunately we have not been able to generate enough new samples to process for qRT from both of these mutants within the revision period. Therefore we have quantitated RNAscope assays across the tumour regions of several mutants of each genotype, facilitated by the auto-fluorescent properties of Fast Red used in this quantitative assay. As expected, the levels of YAP transcriptional output are lower in the YAP-TetO model compared to the LATS1/2 deletion in the *Sox2CreERT2* model. We have included a graph of the quantitation as Figure 5—figure supplement 1E and have included these findings in the Results.

Conceptually, we agree with the reviewers and did not mean to imply that tumours in the LATS1/2 deletion models are due to the intrinsic oncogenic potential of YAP/TAZ. In fact, we are fascinated by the possibility that the strength and duration of YAP/TAZ signaling are the only factors regulating a switch between normal stem cell expansion and tumourigenesis in this setting. Pragmatically, we interpret that results can be due to: (i) enhanced YAP/TAZ activity compared to the YAP-TetO model, as suggested; (ii) additional target expression unique to an increase in TAZ activation, not activated when YAP activity alone is enhanced; or, (iii) abnormal expression of additional, unidentified target proteins of LATS1 that might have oncogenic functions.

We have discussed the differences in levels of signaling between the models in the Discussion section. We have also amended the model in Figure 6 to highlight the difference in the levels of YAP/TAZ target activation.

4) Are the controls displayed in Figure 3—figure supplememt 1F normal tissue from wild-type littermate controls? Would the authors expect them to express markers of adenoma (Synaptophysin, NSE, Chromogranin) so strongly?

Indeed, the controls are from wild-type littermates. We do expect normal endocrine cells of the pituitary to express these neuroendocrine markers strongly (e.g. Stefaneanu L., et al. (1988) Arch Pathol Lab Med. PMID: 3134875; Van Noorden S., et al. (1984) Neuroendocrinology PMID: 6374489; Wilson B.S. and Lloyd R.V. (1984) Am J Pathol. PMID: 6375394). These markers are not diagnostic for pituitary tumour tissue, but are important for the classification of the tumour subtype, since they indicate the neuroendocrine nature of the tumour cells. All adenomas would be positive for Synaptophysin/NSE/Chromogranin, even null-cell adenomas that do not express lineage-commitment transcription factors. The negativity of the tissue is in line with the uncommitted/SOX2+ composition of the tumour, which is not undergoing neuroendocrine commitment as seen in the normal pituitary.

We have clarified that these three are neuroendocrine markers and not markers of adenomas, but that they stain pituitary adenomas, in the last paragraph of the subsection “Loss of LATS kinases results in carcinoma-like murine tumours” (Results).

5) It is not clear why the authors restrict the analysis to embryonic stage in Hesx1^CreER^;R26^rtTA^;YAP^teto^ mice after Dox induction at e5.5. Are they embryonic lethal? If not, what are the postnatal phenotypes in these animals?

We particularly thank the reviewers for this suggestion, as it has revealed a novel phenotype.

Unfortunately with the induction at 5.5dpc, we could not recover any mutants after birth, and have now stated so in the text. However, we adapted the treatment to initiate doxycycline from 9.5dpc and obtained surviving mutants by P24 (4/5). The mutant animals had lower body weight and the anterior pituitary was smaller. Histological examination revealed a phenotype resembling Rathke’s cleft cyst, with multiple cysts developing across all mutant pituitaries. These most likely result from an expansion of SOX2+ cells that maintain epithelial integrity, and although we observe postnatal cell lineage commitment, this is impaired, resulting in in a decrease in PIT1+ cells.

Taking our embryonic and postnatal data together, we therefore conclude that a moderate increase in YAP/TAZ target expression maintains and promotes the SOX2+ fate, translating into deregulated lineage commitment and epithelial expansion with variable phenotypes depending on the timing, while a further elevation leads to the uncontrollable expansion and tumourigenesis. The new data are presented as new Figure1—figure supplement 1B-G, (Results subsection “Sustained conditional expression of YAP during development promotes SOX2+ PSC fate”) and (Discussion).